# A new convergent variant of $Q$-learning with linear function approximation

**Diogo S. Carvalho**      **Francisco S. Melo**      **Pedro A. Santos**
INESC-ID & Instituto Superior Técnico, University of Lisbon
Lisbon, Portugal
`{diogo.s.carvalho, pedro.santos}@tecnico.ulisboa.pt`
`fmelo@inesc-id.pt`

## Abstract

In this work, we identify a novel set of conditions that ensure convergence with probability 1 of $Q$-learning with linear function approximation, by proposing a two time-scale variation thereof. In the faster time scale, the algorithm features an update similar to that of DQN, where the impact of bootstrapping is attenuated by using a $Q$-value estimate akin to that of the target network in DQN. The slower time-scale, in turn, can be seen as a modified target network update. We establish the convergence of our algorithm, provide an error bound and discuss our results in light of existing convergence results on reinforcement learning with function approximation. Finally, we illustrate the convergent behavior of our method in domains where standard $Q$-learning has previously been shown to diverge.

## 1   Introduction

In this paper, we investigate the convergence of reinforcement learning with linear function approximation in control settings. Specifically, we analyze the convergence of $Q$-learning when combined with linear function approximation. Several well-known counter-examples exist in the literature that showcase the divergence of this algorithm when used with even such a relatively "benign" form of function approximation [2, 6, 21]. The divergent behavior has been blamed on the so-called "deadly triad" [18, 24]—function approximation, bootstrapping and off-policy learning. Bootstrapping means that successive estimates for the $Q$-function are built on previous estimates; over-estimation errors for action values thus critically propagate across iterations [23]. Off-policy means that the policy used to sample the environment differs from that which the algorithm is evaluating.

The few results that establish the convergence of $Q$-learning with function approximation either restrict the approximation architecture, eventually minimizing the impact of over-estimation errors [20, 21] or require a very restrictive coupling between the approximation architecture and the sampling distribution which, in practice, occurs only when the sampling policy is very close to the optimal policy, rendering $Q$-learning almost on-policy [13]. More recently, $Q$-learning was attributed finite-time error bounds when certain fixed behaviour policies are used [8]. Unfortunately, such policies are scarce or may not even exist as the number of features grows.

Other convergence results for the control problem in RL with function approximation propose algorithms in which at least one of the elements in the "deadly triad" is not present. For example, in a work of Perkins and Precup (2003), the authors propose a novel algorithm that converges with arbitrary function approximation, but is restricted to on-policy sampling. *Greedy-GQ* [12] is a variant of $Q$-learning with convergence guarantees. However, the associated solution may not be globally optimal, following from the fact it minimizes a non-convex objective function. Finite-time error bounds for Asynchronous Dynamic Programming methods [3], including *Fitted Q-iteration* (FQI)

[9], assume not only realizability of the optimal $Q$-function but also closedness under Bellman update [19, 7]. Finally, convergence results have been established for the return-base setting [16].

Our work is motivated by the success of DQN [14], which can be viewed as an instance of FQI [25]. In their work, Mnih et al. explore two important ideas in order to circumvent (or, at least, mitigate) the negative impact of bootstrapping and off-policy sampling:

- The use of a *target network* to compute the target value for the updates. In the original version, the target network is updated only rarely, by copying the values of the original network, although posterior implementations have adopted Polyak updates, in this sense bringing DQN closer to a two time-scale update scheme;
- The use of *experience replay*, where the samples are drawn from a *replay buffer*, thus minimizing the correlation between samples observed in trajectory-based learning and enabling the use of supervised learning techniques that assume sample independence.

Building on these ideas, we propose a two time-scale variation of $Q$-learning with linear function approximation. Our proposed algorithm keeps two sets of parameters.

- The first set of parameters, corresponding to the "main" iteration, follows a faster time-scale and uses a DQN-like update, where the targets are built from the second set of parameters—thus minimizing the impact of bootstrapping.
- The second set of parameters, corresponding to the "target network", proceeds at a slower time-scale—in a sense mimicking the slower updates of a target network. However, unlike DQN, the second set of parameters does not directly copy the "main" but, instead, a transformed version thereof, reminiscent of the preconditioning process discussed in [1].

We contribute with a convergence analysis of the resulting algorithm, showing convergence *with probability 1*, or *w.p.1*, with much less stringent assumptions than previous works [13] and provide an interpretation and performance bounds for the resulting limit solution.

**Notation:**

We denote random variables (r.v.s) using upright letters, as in x or a, and instances of r.v.s as slanted letters, as in $x$ or $a$. We use uppercase letters to denote functions, as in $V$ or $Q$, and calligraphic letters to denote sets, as in $\mathcal{X}$ or $\mathcal{A}$. Vectors are represented as bold lowercase letters. For example, $\mathbf{c}$ denotes a random vector and $\boldsymbol{c}$ an instance thereof. Matrices are represented using bold uppercase letters, as in $\boldsymbol{Q}$. We write $\mathbb{E}_{x,a \sim p}\left[f(x, a)\right]$ or simply $\mathbb{E}_p\left[f(x, a)\right]$ to denote the expectation of $f$ when the r.v.s x and a follow distribution $p$.

## 2 Background

A Markov decision problem (MDP) is a tuple $(\mathcal{X}, \mathcal{A}, \{\mathbf{P}_a\}, R, \gamma)$, where $\mathcal{X}$ is the countable state space, $\mathcal{A}$ is the finite action space, $\mathbf{P}_a$ is the *transition probability matrix* associated with action $a \in \mathcal{A}$, with component $xx'$ given by

$$[\mathbf{P}_a]_{xx'} \overset{\text{def}}{=} \mathbb{P}\left[x_{t+1} = x' \mid x_t = x, a_t = a\right].$$

The random variable (r.v.) $x_t$ denotes the *state* of the MDP at time step $t$; similarly, the r.v. $a_t$ denotes the *action* of the agent at time step $t$. We generally write $\mathbf{P}(x' \mid x, a)$ to denote $[\mathbf{P}_a]_{xx'}$.

The function $R : \mathcal{X} \times \mathcal{A} \to \mathbb{R}$ is the *expected reward* for performing action $a$ in state $x$. We write $r_t$ to denote the reward at time step $t$. It is a r.v. with expected value $R(x_t, a_t)$ and we assume throughout that $|r_t| \leq \rho$ for some value $\rho > 0$. Finally, $\gamma$ is a discount factor taking values in $[0, 1)$.

Solving an MDP consists in finding a *policy* (i.e., a decision rule) that yields the maximal *total discounted reward*. A policy is a (possibly stochastic) mapping $\pi : \mathcal{X} \to \mathcal{A}$, where $\pi(x)$ is the action selected by $\pi$ at state $x$. The total discounted reward associated with a policy $\pi$ is

$$V^\pi(x) = \mathbb{E}_\pi\left[\sum_{t=0}^\infty \gamma^t r_t \mid x_0 = x\right] = \mathbb{E}_\pi\left[\sum_{t=0}^\infty \gamma^t R(x_t, a_t) \mid x_0 = x\right].$$

We refer to $V^\pi$ as the *value function* associated with policy $\pi$. The optimal policy $\pi^*$ is such that $V^{\pi^*}(x) \geq V^\pi(x)$ for any other policy $\pi$. We write $V^*$ to compactly denote $V^{\pi^*}$ and refer to $V^*$ as the *optimal value function*. The optimal value function verifies the recursive relation

$$V^*(x) = \max_{a \in \mathcal{A}} \left[ R(x, a) + \gamma \sum_{x' \in \mathcal{X}} \mathbf{P}(x' \mid x, a) V^*(x') \right], \tag{1}$$

where the quantity in square brackets is known as the *optimal action-value for the state-action pair* $(x, a)$, and is denoted as $Q^*(x, a)$. Since $V^*(x) = \max_{a \in \mathcal{A}} Q^*(x, a)$, it holds that $Q^*$ verifies

$$Q^*(x, a) = R(x, a) + \gamma \sum_{x' \in \mathcal{X}} \mathbf{P}(x' \mid x, a) \max_{a' \in \mathcal{A}} Q^*(x', a'). \tag{2}$$

An optimal policy is any policy $\pi^*$ such that $\pi^*(x) \in \arg\max_{a \in \mathcal{A}} Q^*(x, a)$.

$V^*$ and $Q^*$ can be computed, respectively, from (1) and (2) using dynamic programming. Alternatively, if $\{\mathbf{P}_a, a \in \mathcal{A}\}$ and $R$ are unknown, they can be computed using stochastic approximation.

In this paper, we are particularly interested in the stochastic approximation approach to the computation of $Q^*$, an algorithm known as *Q-learning*. Given a sequence of observed tuples $\mathcal{B} = \{(x_0, a_0, r_0, x'_0), (x_1, a_1, r_1, x'_1), \ldots, (x_t, a_t, r_t, x'_t), \ldots\}$, obtained by running some learning policy in the MDP, $Q$-learning proceeds by performing, at each time step $t$, the update

$$Q_{t+1}(x_t, a_t) \leftarrow Q_t(x_t, a_t) + \alpha_t \delta_t,$$

where $\delta_t$ is the *temporal difference at time step $t$*, given by

$$\delta_t = r_t + \gamma \max_{a' \in \mathcal{A}} Q_t(x'_t, a') - Q_t(x_t, a_t).$$

Convergence of $Q$-learning can be established using standard stochastic approximation arguments.

## 2.1 $Q$-learning with function approximation

We now address the problem of control in *reinforcement learning with function approximation*, where the optimal $Q$-function, $Q^*$, cannot be represented exactly and, therefore, some form of approximation must be used.

Let us consider a parameterized family of functions $\mathcal{Q} = \{Q_{\boldsymbol{w}}, \boldsymbol{w} \in \mathbb{R}^K\}$, with $Q_{\boldsymbol{w}} : \mathcal{X} \times \mathcal{A} \to \mathbb{R}$. The extension of $Q$-learning to accommodate for one such representation takes the general form

$$\boldsymbol{w}_{t+1} \leftarrow \boldsymbol{w}_t + \alpha_t \nabla Q_{\boldsymbol{w}_t}(x_t, a_t) \delta_t. \tag{3}$$

The update (3) can be seen as a single-sample stochastic gradient update over the error $\mathcal{E} = \mathbb{E}\left[\delta_t^2\right]$, if we assume the target value $R(x_t, a_t) + \gamma \max_{a' \in \mathcal{A}} Q_{\boldsymbol{w}_t}(x'_t, a')$ is fixed. Intuitively, the error $\mathcal{E}$ conveys the notion that the parameters should be adjusted so that the output, $Q_{\boldsymbol{w}_t}(x_t, a_t)$, approaches such target value as well as possible.

Unfortunately, it is a well known phenomenon that having the target value, $R(x_t, a_t) + \gamma \max_{a' \in \mathcal{A}} Q_{\boldsymbol{w}_t}(x'_t, a)$, built from the output $Q_{\boldsymbol{w}_t}$ of the learning algorithm (bootstrapping) may cause the resulting algorithm to diverge.

The recently proposed DQN architecture [14] seeks to alleviate the potential negative impact of bootstrapping by using a *target network* to construct the value of the target above. Such target network is updated infrequently; the targets used to train DQN are, therefore, mostly static. Building on that idea, we contribute a novel method which can be seen as implementing a two time-scale equivalent to the target network in DQN. We analyze the convergence of our proposed algorithm when used with linear function approximators, and contribute with:

- A novel proof of convergence of $Q$-learning with linear function approximation that requires significantly less stringent conditions that those currently available in the literature;

- A better theoretical understanding for the use of the target network in DQN.

# 3 Coupled $Q$-learning

We are given a set of basis functions $\{\phi_1, \ldots, \phi_K\}$, where $\phi_k : \mathcal{X} \times \mathcal{A} \to \mathbb{R}$ for $k = 1, \ldots, K$. The function $Q^*$ is then approximated by a function $Q_{\boldsymbol{w}} \in \{\boldsymbol{\phi} \cdot \boldsymbol{w}, \boldsymbol{w} \in \mathbb{R}^K\}$, where $\boldsymbol{\phi} = (\phi_1, \ldots, \phi_K)$. We designate our method as *coupled $Q$-learning*, or *CQL*, as it consists of the two coupled updates

$$\boldsymbol{u}_{t+1} \leftarrow \boldsymbol{u}_t + \alpha_t \big(\boldsymbol{\phi}(x_t, a_t) Q_{\boldsymbol{v}_t}(x_t, a_t) - \boldsymbol{u}_t\big), \tag{4a}$$

$$\boldsymbol{v}_{t+1} \leftarrow \boldsymbol{v}_t + \beta_t \boldsymbol{\phi}(x_t, a_t) \delta_t. \tag{4b}$$

We now use the temporal difference

$$\delta_t = r_t + \gamma \max_{a' \in \mathcal{A}} Q_{\boldsymbol{u}_t}(x_t', a') - Q_{\boldsymbol{v}_t}(x_t, a_t)$$

and $\alpha_t << \beta_t$. In (4), $Q_{\boldsymbol{u}}$ plays the role of target network and $Q_{\boldsymbol{v}}$ implements the role of the "main" network, implementing the same $Q$-learning update used in DQN. The update (4a) takes place at a slower time scale than the update (4b), emulating the infrequent updates in the target network of DQN [14]. Finally, we do not directly copy the values of the "main" network, but instead match the projection of the output along the feature space, much like the "pre-conditioning" step from [1].

## 3.1 Convergence analysis

We now analyze the convergence of our proposed algorithm. We establish convergence *w.p.1* under the following assumptions.

**(I)** For all $t$, $(x_t, a_t, x_t', r_t)$ is independently sampled from a *replay buffer* $\mathcal{B} = \{(x_i, a_i, x_i', r_i), i \in \mathbb{N}_0\}$ according to a fixed distribution $\mu$. Each tuple $(x_i, a_i, x_i', r_i)$ is sampled from the MDP, i.e., $x_i'$ is distributed according to $\mathbf{P}(\cdot \mid x_i, a_i)$, and $r_i$ is such that $\mathbb{E}_\mu [r_i \mid x_i, a_i] = R(x_i, a_i)$.

**(II)** $\boldsymbol{\Sigma}_\mu \stackrel{\text{def}}{=} \mathbb{E}_\mu \left[\boldsymbol{\phi}(x_t, a_t) \boldsymbol{\phi}^T(x_t, a_t)\right]$ is non-singular and $\|\boldsymbol{\phi}(x, a)\|_2 \leq 1$, for all $(x, a)$.

**(III)** The step size sequences $\{\alpha_t, t \in \mathbb{N}\}$ and $\{\beta_t, t \in \mathbb{N}\}$ verify the conditions $\sum_{t=0}^\infty \alpha_t = \sum_{t=0}^\infty \beta_t = \infty$ and $\sum_{t=0}^\infty \alpha_t^2 + \sum_{t=0}^\infty \beta_t^2 < \infty$. Additionally, $\alpha_t = \text{o}(\beta_t)$.

Our main result follows.

**Theorem 1.** *Under Assumptions (I) through (III), the CQL algorithm defined by the updates* (4) *converges w.p.1.*

Before moving to the proof of Theorem 1, let us consider the implications of Assumptions (I)-(III).

Assumption (I) is similar to the setting considered by Chen and Jiang [7]. It indicates that the tuples $(x_t, a_t, r_t, x_t')$ used to perform the updates are mutually independent—which can be implemented, for example, through a *replay buffer* similar to what is done in many current deep RL approaches. Several previous works have considered the distribution $\mu$ to be the stationary distribution for the Markov chain induced by the sampling policy [13, 20], which typically requires the sampling policy to induce an ergodic Markov chain. In this sense, Assumption (I) is less restrictive, as it makes no particular assumption on the sampling policy but simplifies the convergence analysis.

Assumption (II) requires $\|\boldsymbol{\phi}(x, a)\|_2 \leq 1$ for all $(x, a) \in \mathcal{X} \times \mathcal{A}$. This is a relatively straightforward condition to ensure by a simple scaling of the features $\phi_1, \ldots, \phi_K$ and is also present in recent related work [26, 8]. Assumption (II) also requires that the matrix $\boldsymbol{\Sigma}_\mu$ is non-singular, a condition which is tantamount to the linear independence requirement found in previous works [22]. Assumption (II) is, therefore, significantly weaker than those imposed, for example, in previous works [13], which seldom (if ever) hold in practice.

Assumption (III) is standard for two time-scale algorithms. In practice, the use of small constant step sizes $\alpha$ and $\beta$ is usual, as long as $\alpha << \beta$.

### 3.1.1 Proof of Theorem 1

*Proof.* We establish Theorem 1 by directly applying the well-established two time-scale stochastic approximation result from Borkar [4, Chapter 6] provided in the supplementary material. To do such

application, it amounts to show that our algorithm satisfies each and every condition of the theorem. We present the main steps of the proof and refer to the supplementary material once more for details.

We start by defining the mean fields $F, G : \mathbb{R}^K \times \mathbb{R}^K \to \mathbb{R}^K$ as

$$F(\boldsymbol{u}_t, \boldsymbol{v}_t) \stackrel{\text{def}}{=} \mathbb{E}_\mu \left[ \boldsymbol{\phi}(\mathrm{x}_t, \mathrm{a}_t) \boldsymbol{\phi}^T(\mathrm{x}_t, \mathrm{a}_t) \right] \boldsymbol{v}_t - \boldsymbol{u}_t,$$

$$G(\boldsymbol{u}_t, \boldsymbol{v}_t) \stackrel{\text{def}}{=} \mathbb{E}_\mu \left[ \boldsymbol{\phi}(\mathrm{x}_t, \mathrm{a}_t) \delta_t \mid \boldsymbol{u}_t, \boldsymbol{v}_t \right].$$

Using $F$ and $G$ above, we also define martingale difference sequences of noise $\{\, \mathbf{m}_t \,\}$ and $\{\, \mathbf{n}_t \,\}$ as

$$\mathbf{m}_{t+1} = \left( \boldsymbol{\phi}(x_t, a_t) \boldsymbol{\phi}^T(\mathrm{x}_t, \mathrm{a}_t) \boldsymbol{v}_t - \boldsymbol{u}_t \right) - F(\boldsymbol{u}_t, \boldsymbol{v}_t)$$
$$\mathbf{n}_{t+1} = \boldsymbol{\phi}(x_t, a_t) \delta_t - G(\boldsymbol{u}_t, \boldsymbol{v}_t).$$

Both $F$ and $G$ are Lipschitz continuous, and letting the sigma-algebra $\mathcal{F}_t = \sigma(\{\, (\boldsymbol{u}_\tau, \boldsymbol{v}_\tau, \mathbf{m}_\tau, \mathbf{n}_\tau), \tau = 0, \dots, t \,\})$, we can show that

$$\mathbb{E} \left[ \|\mathbf{m}_{t+1}\|^2 \mid \mathcal{F}_t \right] \le c_{\boldsymbol{m}} (1 + \|\boldsymbol{u}_t\|^2 + \|\boldsymbol{v}_t\|^2),$$

$$\mathbb{E} \left[ \|\mathbf{n}_{t+1}\|^2 \mid \mathcal{F}_t \right] \le c_{\boldsymbol{n}} (1 + \|\boldsymbol{u}_t\|^2 + \|\boldsymbol{v}_t\|^2),$$

for some constants $c_{\boldsymbol{m}}, c_{\boldsymbol{n}} > 0$.

In analyzing two time-scale algorithms we follow the standard notion that, since $\alpha_t \ll \beta_t$, the updates to $\boldsymbol{v}_t$ proceeds at a "faster" timescale than those to $\boldsymbol{u}_t$. Thus, when viewed from the faster time-scale, $\boldsymbol{u}_t$ appears to be *quasi-static*. The update for $\boldsymbol{u}_t$ takes the general form $\boldsymbol{u}_{t+1} = \boldsymbol{u}_t + \beta_t \Delta \boldsymbol{u}_t$, where

$$\Delta \boldsymbol{u}_t \stackrel{\text{def}}{=} \frac{\alpha_t}{\beta_t} \left( \boldsymbol{\phi}(x_t, a_t) \boldsymbol{\phi}^T(x_t, a_t) \boldsymbol{v}_t - \boldsymbol{u}_t \right) \to 0$$

as long as $\boldsymbol{v}_t$ and $\boldsymbol{u}_t$ remain bounded. With this in mind, for a fixed $\boldsymbol{u} \in \mathbb{R}^K$, we have the o.d.e.

$$\dot{\boldsymbol{v}}_t = G(\boldsymbol{u}, \boldsymbol{v}_t),$$

which has a unique globally asymptotically stable equilibrium

$$\boldsymbol{v}^* = \boldsymbol{\lambda}(\boldsymbol{u}) = \boldsymbol{\Sigma}_\mu^{-1} \mathbb{E}_\mu \left[ \boldsymbol{\phi}(\mathrm{x}_t, \mathrm{a}_t) \left( \mathrm{r}_t + \gamma \max_{a' \in \mathcal{A}} \boldsymbol{\phi}^T(\mathrm{x}_t', a') \boldsymbol{u} \right) \mid \boldsymbol{u} \right].$$

The global asymptotic stability of $\boldsymbol{v}^*$ can be established by a Lyapunov argument, using the Lyapunov function $L(\boldsymbol{v}) = \frac{1}{2} \|\boldsymbol{v} - \boldsymbol{v}^*\|^2$. Additionally, $\boldsymbol{\lambda}(\boldsymbol{u})$ can also be shown to be Lipschitz continuous. Finally, defining

$$G_\infty(\boldsymbol{v}) \stackrel{\text{def}}{=} \lim_{c \to \infty} \frac{G(\boldsymbol{u}, c\boldsymbol{v})}{c} = -\boldsymbol{\Sigma}_\mu \boldsymbol{v},$$

we can show that the origin is an asymptotically stable equilibrium for the o.d.e. $\dot{\boldsymbol{v}}_t = G_\infty(\boldsymbol{v}_t)$. From Theorem 2.1 of Borkar and Meyn (2000), we conclude that $\sup_t \|\boldsymbol{v}_t\| < \infty$ *w.p.1.*

Conversely, when viewed from the slower time-scale, $\boldsymbol{v}_t$ appears to have already reached its *equilibrium point*. With this in mind, we have the o.d.e. $\dot{\boldsymbol{u}}_t = F(\boldsymbol{u}_t, \boldsymbol{\lambda}(\boldsymbol{u}_t))$, which also has a unique globally asymptotically stable equilibrium

$$\boldsymbol{u}^* = \mathbb{E}_\mu \left[ \boldsymbol{\phi}(\mathrm{x}_t, \mathrm{a}_t) \left( \mathrm{r}_t + \gamma \max_{a' \in \mathcal{A}} \boldsymbol{\phi}^T(\mathrm{x}_t', a') \boldsymbol{u}^* \right) \mid \boldsymbol{u}^* \right]. \tag{5}$$

The existence of the fixed point in (5) can be established from the Banach fixed-point theorem, since the right-hand side is a contraction in $\boldsymbol{u}$; that $\boldsymbol{u}^*$ is globally asymptotically stable can again be established by a Lyapunov argument, using the function $L(\boldsymbol{u}) = \frac{1}{2} \|\boldsymbol{u} - \boldsymbol{u}^*\|^2$. Finally, we define

$$F_\infty(\boldsymbol{u}) \stackrel{\text{def}}{=} \lim_{c \to \infty} \frac{F(c\boldsymbol{u}, \boldsymbol{\lambda}(c\boldsymbol{u}))}{c} = \mathbb{E}_\mu \left[ \boldsymbol{\phi}(\mathrm{x}_t, \mathrm{a}_t) \gamma \max_{a' \in \mathcal{A}} \boldsymbol{\phi}^T(\mathrm{x}_t, a') \boldsymbol{u} \mid \boldsymbol{u} \right] - \boldsymbol{u}.$$

We can show that the origin is an asymptotically stable equilibrium for the o.d.e. $\dot{\boldsymbol{u}}_t = F_\infty(\boldsymbol{u}_t)$, from where we repeat our previous argument to conclude that $\sup_t \|\boldsymbol{u}_t\| < \infty$ *w.p.1.* Since all conditions have been verified, the conclusion follows. $\qquad \square$

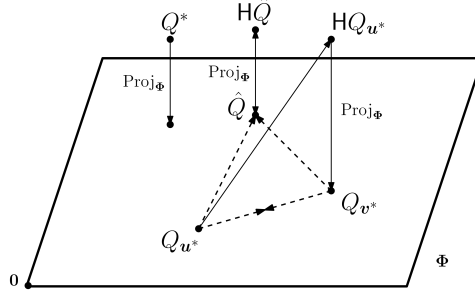

Figure 1: Relation between $\hat{Q}$, $Q^*$ and the coupled solutions $Q_{\boldsymbol{u}^*}$ and $Q_{\boldsymbol{v}^*}$.

## 3.2 Performance analysis

While Theorem 1 establishes the convergence of our algorithm w.p.1, it says nothing about the corresponding limit point. The limit considered in the context of linear function approximation is

$$\hat{Q}(x,a) = (\mathrm{Proj}_\Phi \mathsf{H}\hat{Q})(x,a) \stackrel{\text{def}}{=} \boldsymbol{\phi}^T(x,a)\boldsymbol{\Sigma}_\mu^{-1}\mathbb{E}_\mu\left[\boldsymbol{\phi}(\mathrm{x},\mathrm{a})\Big((\mathsf{H}\hat{Q})(\mathrm{x},\mathrm{a})\Big)\right],$$

where $\mathrm{Proj}_\Phi$ is the orthogonal projection into the span of $\{\,\phi_k : k = 1, \ldots, K\,\}$, which we denote by $\Phi$, and $\mathsf{H}$ is the Bellman operator. In our case we have, instead,

$$Q_{\boldsymbol{u}^*}(x,a) = \boldsymbol{\phi}^T(x,a)\mathbb{E}_\mu\left[\boldsymbol{\phi}(\mathrm{x},\mathrm{a})\Big((\mathsf{H}Q_{\boldsymbol{u}^*})(\mathrm{x},\mathrm{a})\Big)\right].$$

The difference is the normalizing term $\boldsymbol{\Sigma}_\mu^{-1}$. Therefore, $Q_{\boldsymbol{u}}$ is the fixed point of the combined operator obtained from an "un-normalized" orthogonal projection and the Bellman operator.

Taking this analysis one step further, and keeping the parallel between our approach and the DQN architecture, it is also interesting to analyze $Q_{\boldsymbol{v}^*}$, which corresponds to the actual output of the learning algorithm. We have

$$Q_{\boldsymbol{v}^*}(x,a) = \boldsymbol{\phi}^T(x,a)\boldsymbol{\Sigma}_\mu^{-1}\mathbb{E}_\mu\left[\boldsymbol{\phi}(\mathrm{x},\mathrm{a})\Big((\mathsf{H}Q_{\boldsymbol{u}^*})(\mathrm{x},\mathrm{a})\Big)\right].$$

The three functions ($\hat{Q}$, $Q_{\boldsymbol{u}^*}$ and $Q_{\boldsymbol{v}^*}$) coincide when $\boldsymbol{\Sigma}_\mu = \boldsymbol{I}$. However, this case does not fall, in general, under Assumption (II). Nevertheless, consider the case where the set of basis functions $\{\,\phi_k\,\}$ is orthogonal (i.e., $\mathbb{E}_\mu[\phi_i\phi_j] = 0$) and uniformly excited (i.e., $\mathbb{E}_\mu\left[\phi_i^2\right] = \mathbb{E}_\mu\left[\phi_j^2\right] = \sigma$) by a factor $\sigma$ taking values in $(0, 1]$. Equivalently, let the following assumption hold.

(IV) $\boldsymbol{\Sigma}_\mu = \sigma\boldsymbol{I}$ for some non-zero $\sigma \leq 1$.

We note that Assumption (IV) does not impose any additional constraint on the features considered, since we can make them orthogonal and scale them to ensure that the latter assumption holds. Under assumptions (I) through (IV), we can now establish a bound on the error obtained when approximating $Q^*$ by $Q_{\boldsymbol{v}^*}$. Consider the infinity norm $\|\cdot\|$ in the space of bounded real-valued functions on $\mathcal{X} \times \mathcal{A}$.

**Theorem 2.** *The limit $\boldsymbol{v}^*$ of the sequence $\{\,\boldsymbol{v}_t, t \in \mathbb{N}\,\}$ generated by the iterations in (4) is such that*

$$\|Q^* - Q_{\boldsymbol{v}^*}\|_\infty \leq \frac{1}{1-\gamma}\|Q^* - \mathrm{Proj}_\Phi Q^*\|_\infty + \xi_\sigma, \quad where \quad \xi_\sigma = \frac{1-\sigma}{\sigma}\frac{\gamma\rho}{(1-\gamma)^2}. \tag{6}$$

Before presenting the proof of Theorem 2, let us discuss its implications. Consider the right-hand side of (6). Intuitively, the first error term only depends on the proximity between $Q^*$ and its best linear approximation for the given sampling policy and choice of features, within a factor of $1 - \gamma$. But, more importantly, as $\sigma$ approaches 1, the second error term $\xi_\sigma$ goes to 0. Additionally, $\sigma$ is a relevant parameter when we consider the relation between the solution $\hat{Q}$ and the coupled solutions $(Q_{\boldsymbol{u}^*}, Q_{\boldsymbol{v}^*})$, as suggested before. In fact, as $\sigma$ converges to 1, both $Q_{\boldsymbol{u}^*}$ and $Q_{\boldsymbol{v}^*}$ converge to $\hat{Q}$. Figure 1 illustrates, geometrically, the interpretation just discussed. Dashed arrows refer to displacement components as $\sigma$ approaches 1.

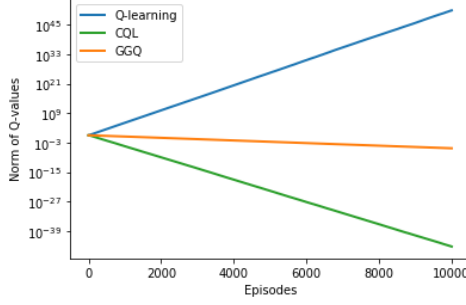 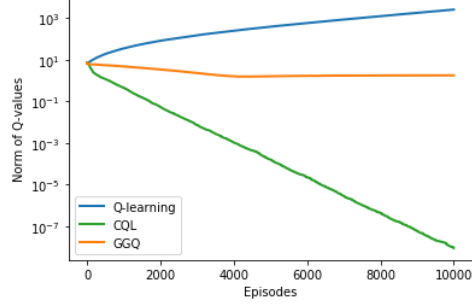

(a) Results on the $\theta \to 2\theta$ example.  (b) Results on the star counterexample.

Figure 2: Comparison of the proposed method with standard $Q$-learning and GGQ.

### 3.2.1 Proof of Theorem 2

*Proof.* We have that
$$\|Q^* - Q_{\boldsymbol{v}^*}\|_\infty \le \|Q^* - \mathrm{Proj}_\Phi Q^*\|_\infty + \|\mathrm{Proj}_\Phi Q^* - Q_{\boldsymbol{v}^*}\|_\infty.$$
Consider the second term on the right-hand side. Since $Q^* = \mathsf{H}Q^*$ and $Q_{\boldsymbol{v}^*} = \mathrm{Proj}_\Phi \mathsf{H}Q_u^*$, we get
$$\|\mathrm{Proj}_\Phi Q^* - Q_{\boldsymbol{v}^*}\|_\infty = \|\mathrm{Proj}_\Phi(\mathsf{H}Q^* - \mathsf{H}Q_{\boldsymbol{u}^*})\|_\infty \le \frac{1}{\sigma}\gamma\|Q^* - Q_{\boldsymbol{u}^*}\|_\infty,$$
by means of the Cauchy-Schwarz and Jensen inequalities and assumption (II).

Since $Q_{\boldsymbol{u}^*} = \sigma Q_{\boldsymbol{v}^*}$ and $\|Q^*\|_\infty \le \rho/(1-\gamma)$, we can put everything together to get
$$\|Q^* - Q_{\boldsymbol{v}^*}\|_\infty \le \|Q^* - \mathrm{Proj}_\Phi Q^*\|_\infty + \gamma\|Q^* - Q_{\boldsymbol{v}^*}\|_\infty + \frac{1-\sigma}{\sigma}\frac{\gamma\rho}{1-\gamma}.$$
Solving the inequality for $\|Q^* - Q_{\boldsymbol{v}^*}\|_\infty$, the desired result follows. □

## 4 Experimental results

We evaluated the CQL algorithm on three domains with increasing complexity. The first was the $\theta \to 2\theta$ example [21] and the second was the 7-star version of the *star counterexample* [2]. Both problems are known two cause divergence of $Q$-learning with linear function approximation. We also tested the algorithm on the *mountain car problem* [15]. On each domain, we compare CQL with standard $Q$-learning and GGQ [12]. We performed *online* learning on the second and third tests, showing that the use of a replay buffer satisfying Assumption (I) is not necessary for convergence.

On the first two domains, results were averaged over 30 runs of $10^3$ episodes, considered $\gamma = 0.99$ and constant learning-rates: $\alpha = 0.1$ for the original algorithm; $\alpha = 0.05, \beta = 0.25$ for CQL and GGQ. After each episode, we compute $\|Q\|_\mathbb{F} = \sum_{x\in\mathcal{X},a\in\mathcal{A}} Q^2(x,a)$.

### 4.1 $\theta \to 2\theta$

In the most simple example [21] there are only two states and one action, and the reward is always zero. The only feature has value 1 for the first state and 2 for the second state. We would expect with the use of a simple architecture such as this one, $Q$-learning would converge to $Q^* = \mathbf{0}$.

We scaled the feature by a factor of $1/2$, set the initial weight as 1, and randomly initialized every episode with equal probability for each state. Each episode consisted of a transition and the update. Figure 2a shows $Q$-learning caused divergence of the $Q$-values, in contrast with the other two methods. The convergence of CQL is considerably faster than of GGQ.

### 4.2 Star counterexample

A more complex domain [2] considers seven states and two actions: the *solid* and the *dotted* action. State seven is absorbing for the solid action, and the dotted action uniformly transitions the agent to any of the first six states. Both actions, on any state, incur zero reward, and thus $Q^* = \mathbf{0}$.

Table 1: Results on the mountain car problem. For each architecture, the best result is bolden.

| Architecture | | Average cumulative reward (± std. dev.) | | |
| $l$ | $\eta$ | $Q$-learning | GGQ | CQL |
|---|---|---|---|---|
| 2 | 0.125 | -190.95 ± 23.42 | **-167.93 ± 39.28** | -172.66 ± 35.82 |
| 2 | 0.25 | -181.67 ± 30.95 | -187.02 ± 24.02 | **-168.03 ± 39.17** |
| 2 | 0.5 | -175.54 ± 35.95 | -180.55 ± 31.08 | **-166.48 ± 38.87** |
| 2 | 1 | -167.98 ± 39.22 | -174.15 ± 36.70 | **-148.99 ± 36.33** |
| 4 | 0.125 | -181.10 ± 31.74 | **-173.75 ± 36.23** | -175.51 ± 34.52 |
| 4 | 0.25 | -174.48 ± 36.57 | **-168.26 ± 34.10** | -175.55 ± 37.35 |
| 4 | 0.5 | -158.12 ± 38.68 | -163.31 ± 35.91 | **-150.59 ± 37.99** |
| 4 | 1 | **-159.80 ± 40.20** | -169.76 ± 35.36 | -164.58 ± 32.55 |
| 8 | 0.125 | -149.12 ± 36.25 | **-133.38 ± 19.25** | -163.42 ± 30.11 |
| 8 | 0.25 | **-121.43 ± 14.50** | -143.83 ± 15.83 | -144.94 ± 33.76 |
| 8 | 0.5 | -181.59 ± 25.35 | **-171.11 ± 33.46** | -182.81 ± 22.97 |
| 8 | 1 | -187.21 ± 20.05 | -193.46 ± 13.60 | **-184.57 ± 15.61** |

We used the same features as those described in the original work [2] but scaled by a factor of $1/\sqrt{5}$. To initiate each run, we set the vectors as $(1, 1, 1, 1, 1, 1, 10, 1)$ for the solid action and the rest as 1 [12]. The auxiliary set of parameters of GGQ was initialized as $\mathbf{0}$. On every run, we initialized the trajectory in any of the six states with equal probability. The dotted action and the solid action were then chosen with probability $5/6$ and $1/6$, respectively, on every episode. Figure 2b shows that the original $Q$-learning caused the Q-values to diverge, whereas CQL and GGQ led them to converge. The GGQ algorithm stabilizes in a sub-optimal solution but CQL converges to the true solution.

### 4.3 Mountain car

The mountain car is a more realistic and complex control problem: a car is placed in the middle of two hills and on each time step it can either accelerate left, do nothing, or accelerate right, forming the action space $\mathcal{A}$. In this environment, gravity is stronger than the engine, and therefore some strategy must be found to climb the hill. Every time step results in a $-1$ reward, except for reaching the goal.

The basis functions used were bi-dimensional Gaussians. Each of the Gaussians had mean in the center of a square on a $l \times l$ grid over the state space $\mathcal{X}$ of position and velocity pairs. The standard deviations were $\sigma_p$ and $\sigma_v$ on the position and velocity dimensions, respectively. The basis functions were normalized so that $\|\phi(x, a)\|_2 = 1$ for each $(x, a)$. Three such functions, in each state, were then associated with the three possible actions. On each run, the initial vectors were $\mathbf{0}$. The learning parameters used were pairs $(\alpha, \beta) \in \{(10^{-i}, 10^{-j}), i = 1, \ldots, 4, j = i, \ldots, 4\}$. In the case of $Q$-learning, only $\alpha$ is used. During training, each run consisted of $10^3$ episodes. Each episode ended when the car successfully climbed the hill or 200 transitions were made. An $\epsilon$-greedy policy was used to learn, with $\epsilon = 0.3$. For testing, after each run, we computed the average cumulative reward of the greedy policy obtained over 100 episodes. Finally, results were averaged over 10 runs.

Table 1 shows, for each algorithm and values of $l$ and $\eta = \frac{\sigma_p}{1.8} = \frac{\sigma_v}{0.14}$, the results obtained from the best learning-rate pair $(\alpha, \beta)$. For CQL, the most selected learning-rate pair was $\alpha = \beta = 10^{-4}$. Also, CQL performed the best when $l = 2$, which testifies for the benefits of simple approximation architectures to the proposed method, as suggested by Theorem 2.

## 5 Conclusions and future work

By proposing a two time-scale variant of $Q$-learning able to combine linear function approximation and off-policy sampling of trajectories, and establishing its convergence under general assumptions, we revived the discussion of convergence for this broadly employed algorithm and introduced a theoretical foundation regarding the use of DQN. We validated the effectiveness of the construction on classical examples where $Q$-learning in its standard configuration fails.

Assumption (II) bounds the feature vectors by the 2-norm. Even though it is achieved without loss of generality, a bound on a non-Euclidean norm, not agnostic to the sampling distribution $\mu$, would be preferable, allowing the limit solutions to be closer to $Q^*$.

## Broader impact

Even though our work is mostly theoretical, we include a reflection on the general impacts of the field.

Artificial intelligence (AI) and machine learning (ML) adoption in society is rapidly increasing. Within the classical application domains—such as robotics, natural language processing, computer vision, predictive models, and others, AI algorithms are now a part of our daily lives. In some of those applications, AI and ML-driven algorithms can surpass human-level performance. Additionally, these algorithms are being used to address critical problems in our world. For example, deep learning algorithms are being used to predict poverty from satellite images [10], and to predict and manage traffic patterns to avoid pollution and congestion in cities [11].

While the impacts of intelligent algorithms are immense, many solid empirical successes are not supported by an equally solid theoretical understanding. This gap between theory and practice is notorious in the field of reinforcement learning (RL), particularly with respect to the recent successes of deep RL. The present work contributes a new algorithm—a modification of $Q$-learning inspired by successful architectures such as DQN—along with the theoretical analysis of its properties. Such contribution has the potential to impact the broader area of RL, including deep RL, by providing a sharper understanding of the theoretical properties of RL algorithms and potentially pushing research towards a new class of stable RL algorithms which make use of more complex approximation architectures than the ones considered in this work—*e.g.*, neural networks.

Given the fundamental nature of the work, we expect its impact on our daily lives to be as far-reaching as that of AI and ML.

## Acknowledgments and Disclosure of Funding

This work was partially supported by national funds through Fundação para a Ciência e Tecnologia under project SLICE with reference PTDC/CCI-COM/30787/2017 and INESC-ID multi annual funding with reference UIDB/50021/2020.

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
