[Supplementary Material]

# Supplementary Material

## A  Two time-scale stochastic approximation result from Borkar [4, Chapter 6]

We provide here a well-established convergence result that we use to establish our main results. The result is taken from a work on stochastic approximation [4] and is used to establish convergence of two time-scale algorithms. Let

$$\mathbf{u}_{t+1} \leftarrow \mathbf{u}_t + \alpha_t\big(F(\boldsymbol{u}_t, \boldsymbol{v}_t) + \mathbf{m}_{t+1}\big) \tag{7a}$$

$$\mathbf{v}_{t+1} \leftarrow \mathbf{v}_t + \beta_t(G(\boldsymbol{u}_t, \boldsymbol{v}_t) + \mathbf{n}_{t+1}), \tag{7b}$$

denote two coupled iterations of a stochastic approximation algorithm, where $\mathbf{u}_t \in \mathbb{R}^p$ and $\mathbf{v}_t \in \mathbb{R}^q$ for all $t$. We consider the following assumptions:

**(A)**  $F : \mathbb{R}^p \times \mathbb{R}^q \to \mathbb{R}^p$ and $G : \mathbb{R}^p \times \mathbb{R}^q \to \mathbb{R}^q$ are both Lipschitz continuous functions;

**(B)**  The step size sequences $\{\,\alpha_t, t \in \mathbb{N}\,\}$ and $\{\,\beta_t, t \in \mathbb{N}\,\}$, are such that

$$\sum_{t=0}^{\infty} \alpha_t = \infty \qquad\qquad \sum_{t=0}^{\infty} \alpha_t^2 < \infty,$$

$$\sum_{t=0}^{\infty} \beta_t = \infty \qquad\qquad \sum_{t=0}^{\infty} \beta_t^2 < \infty,$$

and $\alpha_t = \mathrm{o}(\beta_t)$.

**(C)**  $\{\,\mathbf{m}_t, t \in \mathbb{N}\,\}$ and $\{\,\mathbf{n}_t, t \in \mathbb{N}\,\}$ are two martingale difference sequences w.r.t. the $\sigma$-algebra $\mathcal{F}_t$ generated by $\{\,(\mathbf{u}_\tau, \mathbf{v}_\tau, \mathbf{m}_\tau, \mathbf{n}_\tau), \tau = 0, \dots, t\,\}$. Furthermore, there exist constants $c_{\mathbf{m}}, c_{\mathbf{n}}$ such that for all $t \geq 0$

$$\mathbb{E}\left[\|\mathbf{m}_{t+1}\|^2 | \mathcal{F}_t\right] \leq c_{\mathbf{m}}(1 + \|\boldsymbol{u}_t\|^2 + \|\boldsymbol{v}_t\|^2),$$

$$\mathbb{E}\left[\|\mathbf{n}_{t+1}\|^2 | \mathcal{F}_t\right] \leq c_{\mathbf{n}}(1 + \|\boldsymbol{u}_t\|^2 + \|\boldsymbol{v}_t\|^2).$$

We then have the following result [4].

**Theorem 3.** *Assume that, for every* $\boldsymbol{u} \in \mathbb{R}^p$, *the ordinary differential equation (o.d.e.)*

$$\dot{\boldsymbol{v}}_t = G(\boldsymbol{u}, \boldsymbol{v}_t)$$

*has a unique, globally asymptotically stable equilibrium* $\boldsymbol{\lambda}(\boldsymbol{u})$, *where* $\lambda : \mathbb{R}^p \to \mathbb{R}^q$ *is Lipschitz continuous. Further assume that the o.d.e.*

$$\dot{\boldsymbol{u}}_t = F\big(\boldsymbol{u}_t, \boldsymbol{\lambda}(\boldsymbol{u}_t)\big)$$

*has a unique, globally asymptotically stable equilibrium* $\boldsymbol{u}^* \in \mathbb{R}^p$. *Then, under Assumptions (A) through (C), the coupled iterations* (7) *converge w.p.1 to* $(\boldsymbol{u}^*, \boldsymbol{\lambda}(\boldsymbol{u}^*))$ *as long as* $\sup_t \|\boldsymbol{u}_t\| < \infty$ *and* $\sup_t \|\boldsymbol{v}_t\| < \infty$ *w.p.1.*

## B  Proof of Theorem 1

This appendix provides a more detailed proof of Theorem 1, carefully establishing each of the technical conditions required for the application of Theorem 3. We establish a number of intermediate results (Propositions 1 through 4) that establish the key properties of the mean fields $F$ and $G$ and the martingale difference sequences $\{\,\mathbf{m}_t\,\}$ and $\{\,\mathbf{n}_t\,\}$ defined in the main text. We then establish, in Propositions 5 and 6, the stable behavior for the relevant o.d.e..

## B.1 Preliminaries

For convenience, we repeat herein the relevant definitions from the main text. Our algorithm is defined by the two coupled updates

$$\boldsymbol{u}_{t+1} \leftarrow \boldsymbol{u}_t + \alpha_t\big(\boldsymbol{\phi}(x_t, a_t)Q_{\boldsymbol{v}_t}(x_t, a_t) - \boldsymbol{u}_t\big), \tag{4a}$$

$$\boldsymbol{v}_{t+1} \leftarrow \boldsymbol{v}_t + \beta_t\boldsymbol{\phi}(x_t, a_t)\delta_t, \tag{4b}$$

where

$$\delta_t = r_t + \gamma \max_{a' \in \mathcal{A}} Q_{\boldsymbol{u}_t}(x'_t, a') - Q_{\boldsymbol{v}_t}(x_t, a_t).$$

We assume that

**(I)** For all $t$, $(x_t, a_t, x'_t, r_t)$ is sampled from $\mathcal{B} = \{\, (x_i, a_i, x'_i, r_i),\ i \in \mathbb{N}_0 \,\}$ according to a fixed distribution $\mu$ over $\mathcal{B}$. Moreover, the next-state distribution $x'$ is such that $\mu(x' \mid x, a) = \mathbf{P}(x' \mid x, a)$ for each $x' \in \mathcal{X}$ and the reward distribution $r$ is such that $\mathbb{E}_\mu[r \mid x, a] = R(x, a)$.

**(II)** The matrix $\mathbb{E}_\mu\left[\boldsymbol{\phi}(x_t, a_t)\boldsymbol{\phi}^T(x_t, a_t)\right]$ is non-singular and $\|\boldsymbol{\phi}(x, a)\|_2 \leq 1$, for all pairs $(x, a) \in \mathcal{X} \times \mathcal{A}$.

**(III)** The step size sequences $\{\, \alpha_t, t \in \mathbb{N} \,\}$ and $\{\, \beta_t, t \in \mathbb{N} \,\}$, verify

$$\sum_{t=0}^{\infty} \alpha_t = \infty, \qquad\qquad \sum_{t=0}^{\infty} \alpha_t^2 < \infty,$$

$$\sum_{t=0}^{\infty} \beta_t = \infty, \qquad\qquad \sum_{t=0}^{\infty} \beta_t^2 < \infty,$$

and, moreover, $\alpha_t = o(\beta_t)$.

We define the mean fields

$$F(\boldsymbol{u}_t, \boldsymbol{v}_t) \overset{\text{def}}{=} \mathbb{E}_\mu\left[\boldsymbol{\phi}(x_t, a_t)\boldsymbol{\phi}^T(x_t, a_t)\right]\boldsymbol{v}_t - \boldsymbol{u}_t, \tag{8a}$$

$$G(\boldsymbol{u}_t, \boldsymbol{v}_t) \overset{\text{def}}{=} \mathbb{E}_\mu\left[\boldsymbol{\phi}(x_t, a_t)\delta_t \mid \boldsymbol{u}_t, \boldsymbol{v}_t\right]. \tag{8b}$$

and the martingale differences

$$\mathbf{m}_{t+1} \overset{\text{def}}{=} \left(\boldsymbol{\phi}(x_t, a_t)\boldsymbol{\phi}^T(x_t, a_t)\boldsymbol{v}_t - \boldsymbol{u}_t\right) - F(\boldsymbol{u}_t, \boldsymbol{v}_t), \tag{9a}$$

$$\mathbf{n}_{t+1} \overset{\text{def}}{=} \boldsymbol{\phi}(x_t, a_t)\delta_t - G(\boldsymbol{u}_t, \boldsymbol{v}_t). \tag{9b}$$

Finally, we consider the $\sigma$-algebra

$$\mathcal{F}_t \overset{\text{def}}{=} \sigma(\{\, (\boldsymbol{u}_\tau, \boldsymbol{v}_\tau, \mathbf{m}_\tau, \mathbf{n}_\tau), \tau = 0, \dots, t \,\}).$$

For the upcoming results, it is important to emphasize that, from Assumption (I), the sequence $(x_t, a_t, r_t, x'_t)$ is i.i.d., generated by a fixed distribution $\mu$ and thus independent from $\mathcal{F}_t$ itself. Unless otherwise noted, $\|\cdot\|$ refers to the standard 2-norm in $\mathbb{R}^K$.

## B.2 Lipschitz continuity of $F$ and $G$

We start by establishing $F$ to be Lipschitz continuous.

**Proposition 1.** *The function $F : \mathbb{R}^K \times \mathbb{R}^K \to \mathbb{R}^K$, defined in* (8a)*, is Lipschitz continuous.*

*Proof.* We constructively show that for some $c_F \geq 0$, and for all $\boldsymbol{u}, \boldsymbol{v}, \boldsymbol{w}, \boldsymbol{z} \in \mathbb{R}^K$,

$$\|F(\boldsymbol{u}, \boldsymbol{v}) - F(\boldsymbol{w}, \boldsymbol{z})\| \leq c_F\|(\boldsymbol{u}, \boldsymbol{v}) - (\boldsymbol{w}, \boldsymbol{z})\|.$$

We have that

$$\|F(\boldsymbol{u}, \boldsymbol{v}) - F(\boldsymbol{w}, \boldsymbol{z})\| \leq \left\|\mathbb{E}_\mu\left[\boldsymbol{\phi}(x_t, a_t)\boldsymbol{\phi}^T(x_t, a_t)\right](\boldsymbol{v} - \boldsymbol{z})\right\| + \|\boldsymbol{u} - \boldsymbol{w}\|$$

$$\leq \left\|\mathbb{E}_\mu\left[\boldsymbol{\phi}(x_t, a_t)\boldsymbol{\phi}^T(x_t, a_t)\right]\right\|\|\boldsymbol{v} - \boldsymbol{z}\| + \|\boldsymbol{u} - \boldsymbol{w}\|$$

where the first inequality follows from the triangle inequality, and the second follows from the Cauchy-Schwarz inequality. Using Jensen's inequality and the fact that $\|\phi(x, a)\|_2 \leq 1$, we have that

$$\left\| \mathbb{E}_\mu \left[ \phi(x_t, a_t) \phi^T(x_t, a_t) \right] \right\| \leq \mathbb{E}_\mu \left[ \left\| \phi(x_t, a_t) \phi^T(x_t, a_t) \right\| \right]$$

$$= \mathbb{E}_\mu \left[ \sup_{\|\boldsymbol{x}\|=1} \left\| \phi(x_t, a_t) \phi^T(x_t, a_t) \boldsymbol{x} \right\| \right]$$

$$\leq \mathbb{E}_\mu \left[ \sup_{\|\boldsymbol{x}\|=1} \|\phi(x_t, a_t)\| \left\| \phi^T(x_t, a_t) \boldsymbol{x} \right\| \right]$$

$$\leq 1.$$

This yields

$$\|F(\boldsymbol{u}, \boldsymbol{v}) - F(\boldsymbol{w}, \boldsymbol{z})\| \leq \|\boldsymbol{v} - \boldsymbol{z}\| + \|\boldsymbol{u} - \boldsymbol{w}\| \leq \sqrt{K} \|(\boldsymbol{u}, \boldsymbol{v}) - (\boldsymbol{w}, \boldsymbol{z})\|,$$

and the proof is complete. $\qquad\square$

We now establish a similar result for $G$.

**Proposition 2.** *The function $G : \mathbb{R}^K \times \mathbb{R}^K \to \mathbb{R}^K$, defined in* (8b)*, is Lipschitz continuous.*

*Proof.* We want to show that, for some $c_G \geq 0$,

$$\|G(\boldsymbol{u}, \boldsymbol{v}) - G(\boldsymbol{w}, \boldsymbol{z})\|_2 \leq c_G \|(\boldsymbol{u}, \boldsymbol{v}) - (\boldsymbol{w}, \boldsymbol{z})\|_2,$$

for any fixed $\boldsymbol{u}, \boldsymbol{v}, \boldsymbol{w}, \boldsymbol{z} \in \mathbb{R}^K$. Following along the lines of the proof of Proposition 1, we get

$$\|G(\boldsymbol{u}, \boldsymbol{v}) - G(\boldsymbol{w}, \boldsymbol{z})\|$$
$$\leq \left\| \mathbb{E}_\mu \left[ \gamma \phi(x_t, a_t) (\max_{a' \in \mathcal{A}} \phi^T(x'_t, a') \boldsymbol{u} - \max_{a' \in \mathcal{A}} \phi^T(x'_t, a') \boldsymbol{w}) \right] \right\| + \left\| \mathbb{E}_\mu \left[ \phi(x_t, a_t) \phi^T(x_t, a_t)(\boldsymbol{v} - \boldsymbol{z}) \right] \right\|$$
$$\leq \gamma \mathbb{E}_\mu \left[ \|\phi(x_t, a_t)\| \left| \max_{a' \in \mathcal{A}} \phi^T(x'_t, a') \boldsymbol{u} - \max_{a' \in \mathcal{A}} \phi^T(x'_t, a') \boldsymbol{w} \right| \right] + \left\| \mathbb{E}_\mu \left[ \phi(x_t, a_t) \phi^T(x_t, a_t) \right] \right\| \|\boldsymbol{v} - \boldsymbol{z}\|$$
$$\leq \gamma \mathbb{E}_\mu \left[ \|\phi(x_t, a_t)\| \max_{a' \in \mathcal{A}} \left| \phi^T(x'_t, a')(\boldsymbol{u} - \boldsymbol{w}) \right| \right] + \|\boldsymbol{v} - \boldsymbol{z}\|$$
$$\leq \gamma \mathbb{E}_\mu \left[ \|\phi(x_t, a_t)\| \max_{a' \in \mathcal{A}} \left\| \phi^T(x'_t, a') \right\| \|(\boldsymbol{u} - \boldsymbol{w})\| \right] + \|\boldsymbol{v} - \boldsymbol{z}\|$$
$$\leq \gamma \mathbb{E}_\mu \left[ \|\phi(x_t, a_t)\| \max_{a' \in \mathcal{A}} \left\| \phi^T(x'_t, a') \right\| \right] \|\boldsymbol{u} - \boldsymbol{w}\| + \|\boldsymbol{v} - \boldsymbol{z}\|$$
$$\leq \gamma \|(\boldsymbol{u}, \boldsymbol{v}) - (\boldsymbol{w}, \boldsymbol{z})\|,$$

and the proof is complete. $\qquad\square$

## B.3 Square integrability of $\{\,m_t\,\}$ and $\{\,n_t\,\}$

We start with the following preliminary result.

**Lemma 4.** *The sequence $\{\,(m_t, \mathcal{F}_t), t \in \mathbb{N}\,\}$ defined in* (9a) *is a martingale difference sequence.*

*Proof.* To verify that $\{\,(m_t, \mathcal{F}_t), t \in \mathbb{N}\,\}$ is a martingale difference sequence, we must verify that the following conditions hold for every $t$:

- $\mathbb{E}\left[\|m_t\|\right] < \infty$;
- $\mathbb{E}\left[m_{t+1} \mid \mathcal{F}_t\right] = 0$.

For the first bullet, we have that

$$\mathbb{E}\left[\|\mathbf{m}_{t+1}\|\right] = \mathbb{E}\left[\left\|\left(\phi(\mathrm{x}_t,\mathrm{a}_t)\phi^T(\mathrm{x}_t,\mathrm{a}_t) - \mathbb{E}_\mu\left[\phi(\mathrm{x}_t,\mathrm{a}_t)\phi^T(\mathrm{x}_t,\mathrm{a}_t)\right]\right)\mathbf{v}_t\right\|\right].^1$$

Using the Cauchy-Schwartz inequality and the linearity of the expectation, we get

$$\mathbb{E}\left[\|\mathbf{m}_{t+1}\|\right] \leq \mathbb{E}\left[\left\|\phi(\mathrm{x}_t,\mathrm{a}_t)\phi^T(\mathrm{x}_t,\mathrm{a}_t) - \mathbb{E}_\mu\left[\phi(\mathrm{x}_t,\mathrm{a}_t)\phi^T(\mathrm{x}_t,\mathrm{a}_t)\right]\right\|\right]\mathbb{E}\left[\|\mathbf{v}_t\|\right]$$

$$\leq \mathbb{E}\left[\left\|\phi(\mathrm{x}_t,\mathrm{a}_t)\phi^T(\mathrm{x}_t,\mathrm{a}_t)\right\| + \left\|\mathbb{E}_\mu\left[\phi(\mathrm{x}_t,\mathrm{a}_t)\phi^T(\mathrm{x}_t,\mathrm{a}_t)\right]\right\|\right]\mathbb{E}\left[\|\mathbf{v}_t\|\right].$$

Repeating the steps from the proof of Proposition 1, it follows that

$$\mathbb{E}\left[\left\|\phi(\mathrm{x}_t,\mathrm{a}_t)\phi^T(\mathrm{x}_t,\mathrm{a}_t)\right\|\right] \leq 1,$$

yielding

$$\mathbb{E}\left[\|\mathbf{m}_{t+1}\|\right] \leq 2\mathbb{E}\left[\|\mathbf{v}_t\|\right] < \infty,$$

since $\mathbf{v}_t$ is constructed by a finite number of algebraic operations over finite quantities.[2]

To finish the proof, it remains to show that $\mathbb{E}\left[\mathbf{m}_{t+1} \mid \mathcal{F}_t\right] = 0$. We have that

$$\mathbb{E}\left[\mathbf{m}_{t+1}|\mathcal{F}_t\right] = \mathbb{E}\left[\left(\phi(\mathrm{x}_t,\mathrm{a}_t)\phi^T(\mathrm{x}_t,\mathrm{a}_t) - \mathbb{E}_\mu\left[\phi(\mathrm{x}_t,\mathrm{a}_t)\phi^T(\mathrm{x}_t,\mathrm{a}_t)\right]\right)\mathbf{v}_t \mid \mathcal{F}_t\right]$$

$$= \mathbb{E}\left[\phi(\mathrm{x}_t,\mathrm{a}_t)\phi^T(\mathrm{x}_t,\mathrm{a}_t) - \mathbb{E}_\mu\left[\phi(\mathrm{x}_t,\mathrm{a}_t)\phi^T(\mathrm{x}_t,\mathrm{a}_t)\right] \mid \mathcal{F}_t\right]\mathbf{v}_t$$

$$= \left(\mathbb{E}\left[\phi(\mathrm{x}_t,\mathrm{a}_t)\phi^T(\mathrm{x}_t,\mathrm{a}_t) \mid \mathcal{F}_t\right] - \mathbb{E}\left[\mathbb{E}_\mu\left[\phi(\mathrm{x}_t,\mathrm{a}_t)\phi^T(\mathrm{x}_t,\mathrm{a}_t)\right] \mid \mathcal{F}_t\right]\right)\mathbf{v}_t$$

$$= \left(\mathbb{E}_\mu\left[\phi(\mathrm{x}_t,\mathrm{a}_t)\phi^T(\mathrm{x}_t,\mathrm{a}_t)\right] - \mathbb{E}_\mu\left[\phi(\mathrm{x}_t,\mathrm{a}_t)\phi^T(\mathrm{x}_t,\mathrm{a}_t)\right]\right)\mathbf{v}_t$$

$$= 0.$$

$\square$

For the sequence $\{\,\mathbf{n}_t\,\}$, we have a similar result.

**Lemma 5.** *The sequence* $\{\,(\mathbf{n}_t,\mathcal{F}_t), t \in \mathbb{N}\,\}$ *defined in* (9b) *is a martingale difference sequence.*

*Proof.* As in the proof of Lemma 4, the following must hold:

- $\mathbb{E}\left[\|\mathbf{n}_t\|\right] < \infty$;
- $\mathbb{E}\left[\mathbf{n}_{t+1} \mid \mathcal{F}_t\right] = 0$.

Writing down the expression for $\mathbf{n}_t$, we get

$$\mathbb{E}\left[\|\mathbf{n}_{t+1}\|\right] = \mathbb{E}\left[\|\phi(\mathrm{x}_t,\mathrm{a}_t)\delta_t - \mathbb{E}_\mu\left[\phi(\mathrm{x}_t,\mathrm{a}_t)\delta_t \mid \boldsymbol{u}_t,\boldsymbol{v}_t\right]\|\right]$$

$$\leq \mathbb{E}\left[\|\phi(\mathrm{x}_t,\mathrm{a}_t)\delta_t\| + \|\mathbb{E}_\mu\left[\phi(\mathrm{x}_t,\mathrm{a}_t)\delta_t \mid \boldsymbol{u}_t,\boldsymbol{v}_t\right]\|\right],$$

where the inequality follows from the triangle inequality. Using Jensen's inequality, we then get

$$\mathbb{E}\left[\|\mathbf{n}_{t+1}\|\right] \leq \mathbb{E}\left[\|\phi(\mathrm{x}_t,\mathrm{a}_t)\delta_t\| + \mathbb{E}_\mu\left[\|\phi(\mathrm{x}_t,\mathrm{a}_t)\delta_t\| \mid \boldsymbol{u}_t,\boldsymbol{v}_t\right]\right] < \infty,$$

where the last inequality follows from the fact that all terms are bounded.

To show that $\mathbb{E}\left[\mathbf{n}_{t+1} \mid \mathcal{F}_t\right] = 0$, we can write

$$\mathbb{E}\left[\mathbf{n}_{t+1}|\mathcal{F}_t\right] = \mathbb{E}\left[\phi(\mathrm{x}_t,\mathrm{a}_t)\delta_t - \mathbb{E}_\mu\left[\phi(\mathrm{x}_t,\mathrm{a}_t)\delta_t \mid \boldsymbol{u}_t,\boldsymbol{v}_t\right] \mid \mathcal{F}_t\right]$$

$$= \mathbb{E}\left[\phi(\mathrm{x}_t,\mathrm{a}_t)\delta_t \mid \mathcal{F}_t\right] - \mathbb{E}\left[\mathbb{E}_\mu\left[\phi(\mathrm{x}_t,\mathrm{a}_t)\delta_t \mid \boldsymbol{u}_t,\boldsymbol{v}_t\right] \mid \mathcal{F}_t\right]$$

$$= \mathbb{E}_\mu\left[\phi(\mathrm{x}_t,\mathrm{a}_t)\delta_t \mid \boldsymbol{u}_t,\boldsymbol{v}_t\right] - \mathbb{E}_\mu\left[\phi(\mathrm{x}_t,\mathrm{a}_t)\delta_t \mid \boldsymbol{u}_t,\boldsymbol{v}_t\right]$$

$$= 0.$$

$\square$

Endowed with Lemmas 4 and 5, we proceed to establishing square-integrability of $\{\mathbf{m}_t\}$ and $\{\mathbf{n}_t\}$.

**Proposition 3.** *There exists $c_{\mathbf{m}} > 0$ such that, for any $t > 0$,*

$$\mathbb{E}\left[\|\mathbf{m}_{t+1}\|^2 \mid \mathcal{F}_t\right] \le c_{\mathbf{m}}(1 + \|\boldsymbol{u}_t\|^2 + \|\boldsymbol{v}_t\|^2).$$

*Proof.* From the definition, we get

$$
\begin{aligned}
\mathbb{E}\left[\|\mathbf{m}_{t+1}\|^2 \mid \mathcal{F}_t\right] &= \mathbb{E}\left[\left\|\left(\boldsymbol{\phi}(x_t, a_t)\boldsymbol{\phi}^T(\mathbf{x}_t, \mathbf{a}_t) - \mathbb{E}_\mu\left[\boldsymbol{\phi}(\mathbf{x}_t, \mathbf{a}_t)\boldsymbol{\phi}^T(\mathbf{x}_t, \mathbf{a}_t)\right]\right)\mathbf{v}_t\right\|^2 \mid \mathcal{F}_t\right] \\
&\le \mathbb{E}\left[\left(\left\|(\boldsymbol{\phi}(x_t, a_t)\boldsymbol{\phi}^T(\mathbf{x}_t, \mathbf{a}_t)\right\| + \left\|\mathbb{E}_\mu\left[\boldsymbol{\phi}(\mathbf{x}_t, \mathbf{a}_t)\boldsymbol{\phi}^T(\mathbf{x}_t, \mathbf{a}_t)\right]\right\|\right)^2 \|\boldsymbol{v}_t\|^2 \mid \mathcal{F}_t\right] \\
&\le 4\|\boldsymbol{v}_t\|^2 \\
&\le 4(1 + \|\boldsymbol{u}_t\|_2^2 + \|\boldsymbol{v}_t\|_2^2).
\end{aligned}
$$

$\square$

For $\{\mathbf{n}_t\}$, we get a similar result.

**Proposition 4.** *There exists $c_{\mathbf{n}} > 0$ such that, for any $t > 0$,*

$$\mathbb{E}\left[\|\mathbf{n}_{t+1}\|^2 \mid \mathcal{F}_t\right] \le c_{\mathbf{n}}(1 + \|\boldsymbol{u}_t\|^2 + \|\boldsymbol{v}_t\|^2).$$

*Proof.* Note that

$$
\begin{aligned}
\mathbb{E}\left[\|\mathbf{n}_{t+1}\|^2 \mid \mathcal{F}_t\right] &= \mathbb{E}\left[\|\boldsymbol{\phi}(\mathbf{x}_t, \mathbf{a}_t)\delta_t - \mathbb{E}_\mu\left[\boldsymbol{\phi}(\mathbf{x}_t, \mathbf{a}_t)\delta_t \mid \boldsymbol{u}_t, \boldsymbol{v}_t\right]\|^2 \mid \mathcal{F}_t\right] \\
&\le \mathbb{E}\left[(\|\boldsymbol{\phi}(\mathbf{x}_t, \mathbf{a}_t)\delta_t\| + \|\mathbb{E}\left[\boldsymbol{\phi}(\mathbf{x}_t, \mathbf{a}_t)\delta_t \mid \boldsymbol{u}_t, \boldsymbol{v}_t\right]\|)^2 \mid \mathcal{F}_t\right] \\
&= 4\mathbb{E}\left[\|\boldsymbol{\phi}(\mathbf{x}_t, \mathbf{a}_t)\delta_t\|^2 \mid \mathcal{F}_t\right].
\end{aligned}
$$

Breaking down the right-hand side, we get

$$
\begin{aligned}
\mathbb{E}\left[\|\boldsymbol{\phi}(\mathbf{x}_t, \mathbf{a}_t)\delta_t\|^2 \mid \mathcal{F}_t\right] &= \mathbb{E}\left[\left\|\boldsymbol{\phi}(\mathbf{x}_t, \mathbf{a}_t)(\mathbf{r}_t + \gamma \max_{a' \in \mathcal{A}} \boldsymbol{\phi}^T(\mathbf{x}_t', a')\boldsymbol{u}_t - \boldsymbol{\phi}^T(\mathbf{x}_t, \mathbf{a}_t)\boldsymbol{v}_t)\right\|^2 \mid \mathcal{F}_t\right] \\
&\le \mathbb{E}\left[\|\boldsymbol{\phi}(\mathbf{x}_t, \mathbf{a}_t)\|^2 \left|\mathbf{r}_t + \gamma \max_{a' \in \mathcal{A}} \boldsymbol{\phi}^T(\mathbf{x}_t', a')\boldsymbol{u}_t - \boldsymbol{\phi}^T(\mathbf{x}_t, \mathbf{a}_t)\boldsymbol{v}_t\right|^2 \mid \mathcal{F}_t\right] \\
&\le \mathbb{E}\left[\rho + \left(\left|\gamma \max_{a' \in \mathcal{A}} \boldsymbol{\phi}^T(\mathbf{x}_t', a')\boldsymbol{u}_t\right| + \left|\boldsymbol{\phi}^T(\mathbf{x}_t, \mathbf{a}_t)\boldsymbol{v}_t\right|\right)^2 \mid \mathcal{F}_t\right] \\
&\le \mathbb{E}\left[\rho + (\gamma \max_{a' \in \mathcal{A}} \left\|\boldsymbol{\phi}^T(\mathbf{x}_t, a')\right\| \|\boldsymbol{u}_t\| + \left\|\boldsymbol{\phi}^T(\mathbf{x}_t, \mathbf{a}_t)\right\| \|\boldsymbol{v}_t\|)^2 \mid \mathcal{F}_t\right] \\
&\le \rho + (\|\boldsymbol{u}_t\| + \|\boldsymbol{v}_t\|)^2 \\
&\le c_{\mathbf{n}}(1 + \|\boldsymbol{u}_t\|^2 + \|\boldsymbol{v}_t\|^2),
\end{aligned}
$$

where $c_{\mathbf{n}}$ depends on $\rho$ and $K$. $\square$

### B.4 Stability of the o.d.e.

Propositions 1 through 4 establish that our algorithm verifies the technical assumptions of Theorem 3. It remains to show that the remaining conditions of theorem also hold—namely, the stability of the associated o.d.e. and the boundedness of the iterates $\boldsymbol{u}_t$ and $\boldsymbol{v}_t$.

**Proposition 5.** *For any fixed $\boldsymbol{u} \in \mathbb{R}^K$, the ordinary differential equation*

$$\dot{\boldsymbol{v}}_t = G(\boldsymbol{u}, \boldsymbol{v}_t) \tag{10}$$

*has a unique, globally asymptotically stable equilibrium $\boldsymbol{\lambda}(\boldsymbol{u})$, where $\boldsymbol{\lambda} : \mathbb{R}^K \to \mathbb{R}^K$ is Lipschitz continuous.*

*Proof.* For a given $\boldsymbol{u}$, $\boldsymbol{v}^* \in \mathbb{R}^K$ is an equilibrium of the o.d.e. (10) if

$$G(\boldsymbol{u}, \boldsymbol{v}^*) = 0$$

or, equivalently, if

$$\boldsymbol{v}^* = \boldsymbol{\Sigma}_\mu^{-1} \mathbb{E}_\mu \left[ \boldsymbol{\phi}(\mathbf{x}_t, \mathbf{a}_t)\big(\mathbf{r}_t + \gamma \max_{a' \in \mathcal{A}} \boldsymbol{\phi}^T(\mathbf{x}'_t, a')\boldsymbol{u}\big) \right].$$

That such equilibrium exists for any $\boldsymbol{u}$ follows from the fact that $\boldsymbol{\Sigma}_\mu$ is non-singular, as required by Assumption (II). Define $\boldsymbol{\lambda} : \mathbb{R}^K \to \mathbb{R}^K$ as

$$\boldsymbol{\lambda}(\boldsymbol{u}) = \boldsymbol{\Sigma}_\mu^{-1} \mathbb{E}_\mu \left[ \boldsymbol{\phi}(\mathbf{x}_t, \mathbf{a}_t)\big(\mathbf{r}_t + \gamma \max_{a' \in \mathcal{A}} \boldsymbol{\phi}^T(\mathbf{x}'_t, a')\boldsymbol{u}\big) \right].$$

To see that the function $\boldsymbol{\lambda}$ thus defined is Lipschitz continuous, we note that

$$\|\boldsymbol{\lambda}(\boldsymbol{u}) - \boldsymbol{\lambda}(\boldsymbol{u}')\| = \left\| \boldsymbol{\Sigma}_\mu^{-1} \mathbb{E}_\mu \left[ \gamma\boldsymbol{\phi}(\mathbf{x}_t, \mathbf{a}_t)(\max_{a' \in \mathcal{A}} \boldsymbol{\phi}^T(\mathbf{x}'_t, a')\boldsymbol{u} - \max_{a' \in \mathcal{A}} \boldsymbol{\phi}^T(\mathbf{x}'_t, a')\boldsymbol{u}') \right] \right\|$$

$$\leq \left\| \boldsymbol{\Sigma}_\mu^{-1} \right\| \left\| \mathbb{E}_\mu \left[ \gamma\boldsymbol{\phi}(\mathbf{x}_t, \mathbf{a}_t)(\max_{a' \in \mathcal{A}} \boldsymbol{\phi}^T(\mathbf{x}'_t, a')\boldsymbol{u} - \max_{a' \in \mathcal{A}} \boldsymbol{\phi}^T(\mathbf{y}_t, a')\boldsymbol{u}') \right] \right\|.$$

Using Jensen's inequality, we get

$$\|\boldsymbol{\lambda}(\boldsymbol{u}) - \boldsymbol{\lambda}(\boldsymbol{u}')\| \leq \left\| \boldsymbol{\Sigma}_\mu^{-1} \right\| \mathbb{E}_\mu \left[ \left\| \gamma\boldsymbol{\phi}(\mathbf{x}_t, \mathbf{a}_t)(\max_{a' \in \mathcal{A}} \boldsymbol{\phi}^T(\mathbf{x}'_t, a')\boldsymbol{u} - \max_{a' \in \mathcal{A}} \boldsymbol{\phi}^T(\mathbf{x}'_t, a')\boldsymbol{u}') \right\| \right]$$

$$\leq \left\| \boldsymbol{\Sigma}_\mu^{-1} \right\| \mathbb{E}_\mu \left[ \gamma\|\boldsymbol{\phi}(\mathbf{x}_t, \mathbf{a}_t)\| \max_{a' \in \mathcal{A}} \left| \boldsymbol{\phi}^T(\mathbf{x}'_t, a')(\boldsymbol{u} - \boldsymbol{u}') \right| \right]$$

$$\leq \left\| \boldsymbol{\Sigma}_\mu^{-1} \right\| \mathbb{E}_\mu \left[ \gamma\|\boldsymbol{\phi}(\mathbf{x}_t, \mathbf{a}_t)\| \max_{a' \in \mathcal{A}} \left\| \boldsymbol{\phi}^T(\mathbf{x}'_t, a') \right\| \right] \|\boldsymbol{u} - \boldsymbol{u}'\|$$

$$\leq c_{\boldsymbol{\lambda}} \|\boldsymbol{u} - \boldsymbol{u}'\|,$$

for some $c_{\boldsymbol{\lambda}} > 0$.

Finally, to show that, given $\boldsymbol{u}$, $\boldsymbol{\lambda}(\boldsymbol{u})$ is a globally asymptotically stable equilibrium for the o.d.e. (10), we define the candidate Lyapunov function $L_G : \mathbb{R}^K \to \mathbb{R}$ as

$$L_G(\boldsymbol{v}) = \frac{1}{2}\|\boldsymbol{v} - \boldsymbol{\lambda}(\boldsymbol{u})\|^2.$$

The proof is complete as long as

1. $L_G(\boldsymbol{v}) \geq 0$ for all $\boldsymbol{v} \in \mathbb{R}^K$;

2. $L_G(\boldsymbol{v}) = 0$ if and only if $\boldsymbol{v} = \boldsymbol{\lambda}(\boldsymbol{u})$;

3. $\dot{L}_G(\boldsymbol{v}) \leq 0$ for all $\boldsymbol{v} \in \mathbb{R}^K$;

4. $\dot{L}_G(\boldsymbol{v}) = 0$ if and only if $\boldsymbol{v} = \boldsymbol{\lambda}(\boldsymbol{u})$.

The first two properties follow directly from the definition of $L_G$. As for the last two, we start by explicitly writing $\dot{L}_G$, to get

$$\dot{L}_G(\boldsymbol{v}) \overset{\text{def}}{=} \frac{d}{dt} L_G(\boldsymbol{v}) = \sum_{k=1}^{K} \frac{\partial L_G}{\partial \boldsymbol{v}_k} G_k(\boldsymbol{u}, \boldsymbol{v}) = (\boldsymbol{v} - \boldsymbol{\lambda}(\boldsymbol{u}))^T G(\boldsymbol{u}, \boldsymbol{v})$$

Hence,

$$\dot{L}_G(\boldsymbol{v}) = (\boldsymbol{v} - \boldsymbol{\lambda}(\boldsymbol{u}))^T \mathbb{E}_\mu \left[ \boldsymbol{\phi}(\mathbf{x}_t, \mathbf{a}_t)(\mathbf{r}_t + \gamma \max_{a' \in A} \boldsymbol{\phi}^T(\mathbf{x}'_t)\boldsymbol{u} - \boldsymbol{\phi}^T(\mathbf{x}_t, \mathbf{a}_t)\boldsymbol{v}) \right]$$

$$= (\boldsymbol{v} - \boldsymbol{\lambda}(\boldsymbol{u}))^T \left( \mathbb{E}_\mu \left[ \boldsymbol{\phi}(\mathbf{x}_t, \mathbf{a}_t)(\mathbf{r}_t + \gamma \max_{a' \in A} \boldsymbol{\phi}^T(\mathbf{x}'_t, a')\boldsymbol{u} - \boldsymbol{\phi}^T(\mathbf{x}_t, \mathbf{a}_t)\boldsymbol{v}) \right] - G(\boldsymbol{u}, \boldsymbol{\lambda}(\boldsymbol{u})) \right),$$

where we used the fact that $G(\boldsymbol{u}, \boldsymbol{\lambda}(\boldsymbol{u})) = 0$. Then, after some shuffling, we finally get

$$= (\boldsymbol{v} - \boldsymbol{\lambda}(\boldsymbol{u}))^T \mathbb{E}_\mu \left[ \boldsymbol{\phi}(\mathrm{x}_t, \mathrm{a}_t) \boldsymbol{\phi}^T (\mathrm{x}_t, \mathrm{a}_t)(\boldsymbol{\lambda}(\boldsymbol{u}) - \boldsymbol{v}) \right]$$

$$= -(\boldsymbol{v} - \boldsymbol{\lambda}(\boldsymbol{u}))^T \Sigma_\mu (\boldsymbol{v} - \boldsymbol{\lambda}(\boldsymbol{u})) \le 0,$$

where the last inequality comes from the fact that $\mathbb{E}_\mu \left[ \boldsymbol{\phi}(\mathrm{x}_t, \mathrm{a}_t) \boldsymbol{\phi}^T (\mathrm{x}_t, \mathrm{a}_t) \right]$ is an auto-covariance matrix and, as such, positive definite. The conclusion follows. □

Next, we present a similar result for the slower o.d.e.

**Proposition 6.** *The ordinary differential equation*

$$\dot{\boldsymbol{u}}_t = F(\boldsymbol{u}_t, \boldsymbol{\lambda}(\boldsymbol{u}_t)) \tag{11}$$

*has a unique, globally asymptotically stable equilibrium $\boldsymbol{u}^* \in \mathbb{R}^K$.*

*Proof.* We start by establishing the existence of at least one equilibrium. A vector $\boldsymbol{u}^* \in \mathbb{R}^K$ is an equilibrium for the o.d.e. (11) if

$$F(\boldsymbol{u}^*, \boldsymbol{\lambda}(\boldsymbol{u}^*)) = 0$$

or, equivalently, if

$$\boldsymbol{u}^* = \mathbb{E}_\mu \left[ \boldsymbol{\phi}(\mathrm{x}_t, \mathrm{a}_t)(\mathrm{r}_t + \gamma \max_{a' \in \mathcal{A}} \boldsymbol{\phi}^T (\mathrm{x}'_t, a') \boldsymbol{u}^*) \right].$$

Define $H' : \mathbb{R}^K \to \mathbb{R}^K$ as

$$H'(\boldsymbol{u}) = \mathbb{E}_\mu \left[ \boldsymbol{\phi}(\mathrm{x}_t, \mathrm{a}_t)(\mathrm{r}_t + \gamma \max_{a' \in \mathcal{A}} \boldsymbol{\phi}^T (\mathrm{x}'_t, a') \boldsymbol{u}) \right].$$

Then, the equilibria for the o.d.e. (11) correspond to the fixed points of $H'$. By showing $H'$ to be a contraction, Banach's fixed point theorem ensures existence of a single fixed point, thus establishing both the existence and unicity of an equilibrium for (11). Given any $\boldsymbol{z}, \boldsymbol{w} \in \mathbb{R}^K$,

$$\| H'(\boldsymbol{w}) - H'(\boldsymbol{z}) \|$$

$$= \left\| \mathbb{E}_\mu \left[ \boldsymbol{\phi}(\mathrm{x}_t, \mathrm{a}_t)(\mathrm{r}_t + \gamma \max_{a' \in \mathcal{A}} \boldsymbol{\phi}^T (\mathrm{x}'_t, a') \boldsymbol{w}) \right] - \mathbb{E}_\mu \left[ \boldsymbol{\phi}(\mathrm{x}_t, \mathrm{a}_t)[\mathrm{r}_t + \gamma \max_{a' \in \mathcal{A}} \boldsymbol{\phi}^T (\mathrm{x}'_t, a') \boldsymbol{z}] \right] \right\|$$

$$= \left\| \mathbb{E}_\mu \left[ \gamma \boldsymbol{\phi}(\mathrm{x}_t, \mathrm{a}_t)(\max_{a' \in \mathcal{A}} \boldsymbol{\phi}^T (\mathrm{x}'_t, a') \boldsymbol{w} - \max_{a' \in \mathcal{A}} \boldsymbol{\phi}^T (\mathrm{x}'_t, a') \boldsymbol{z}) \right] \right\|.$$

Using Jensen's inequality,

$$\| H(\boldsymbol{w}) - H(\boldsymbol{z}) \| \le \mathbb{E}_\mu \left[ \left\| \gamma \boldsymbol{\phi}(\mathrm{x}_t, \mathrm{a}_t)(\max_{a' \in \mathcal{A}} \boldsymbol{\phi}^T (\mathrm{x}'_t, a') \boldsymbol{w} - \max_{a' \in \mathcal{A}} \boldsymbol{\phi}^T (\mathrm{x}'_t, a') \boldsymbol{z}) \right\| \right]$$

$$= \mathbb{E}_\mu \left[ \gamma \| \boldsymbol{\phi}(\mathrm{x}_t, \mathrm{a}_t) \| \left| \max_{a' \in \mathcal{A}} \boldsymbol{\phi}^T (\mathrm{x}'_t, a') \boldsymbol{w} - \max_{a' \in \mathcal{A}} \boldsymbol{\phi}^T (\mathrm{x}'_t, a') \boldsymbol{z} \right| \right]$$

$$\le \mathbb{E}_\mu \left[ \gamma \| \boldsymbol{\phi}(\mathrm{x}_t, \mathrm{a}_t) \| \max_{a' \in \mathcal{A}} \left| \boldsymbol{\phi}^T (\mathrm{x}'_t, a') \boldsymbol{w} - \boldsymbol{\phi}^T (\mathrm{x}'_t, a') \boldsymbol{z} \right| \right]$$

$$= \mathbb{E}_\mu \left[ \gamma \| \boldsymbol{\phi}(\mathrm{x}_t, \mathrm{a}_t) \| \max_{a' \in \mathcal{A}} \left| \boldsymbol{\phi}^T (\mathrm{x}'_t, a')(\boldsymbol{w} - \boldsymbol{z}) \right| \right]$$

$$\le \mathbb{E}_\mu \left[ \gamma \| \boldsymbol{\phi}(\mathrm{x}_t, \mathrm{a}_t) \| \max_{a' \in \mathcal{A}} \left\| \boldsymbol{\phi}^T (\mathrm{x}'_t, a') \right\| \| \boldsymbol{w} - \boldsymbol{z} \| \right]$$

$$\le \gamma \| \boldsymbol{w} - \boldsymbol{z} \|.$$

It follows that there is, in fact, a unique equilibrium $\boldsymbol{u}^*$ for the o.d.e. To show that it is globally asymptotically stable, we again use Lyapunov's second method. We define the candidate Lyapunov function $L_F : \mathbb{R}^K \to \mathbb{R}$ as

$$L_F(\boldsymbol{u}) = \frac{1}{2} \| \boldsymbol{u} - \boldsymbol{u}^* \|^2.$$

Once again, the conclusion follows as long as

1. $L_F(\boldsymbol{u}) \geq 0$ for all $\boldsymbol{u} \in \mathbb{R}^K$;

2. $L_F(\boldsymbol{u}) = 0$ if and only if $\boldsymbol{u} = \boldsymbol{u}^*$;

3. $\dot{L}_F(\boldsymbol{u}) \leq 0$ for all $\boldsymbol{u} \in \mathbb{R}^K$;

4. $\dot{L}_F(\boldsymbol{u}) = 0$ if and only if $\boldsymbol{u} = \boldsymbol{u}^*$.

The first two follow directly from the definition of $L_F$. As for the last two,

$$\dot{L}_F(\boldsymbol{u}) = (\boldsymbol{u} - \boldsymbol{u}^*)^T F(\boldsymbol{u}, \boldsymbol{\lambda}(\boldsymbol{u}))^T$$
$$= (\boldsymbol{u} - \boldsymbol{u}^*)^T \mathbb{E}_\mu \left[ \gamma \boldsymbol{\phi}(\mathrm{x}_t, \mathrm{a}_t)(\max_{a' \in \mathcal{A}} \boldsymbol{\phi}^T(\mathrm{x}'_t, a')\boldsymbol{u} - \max_{a' \in \mathcal{A}} \boldsymbol{\phi}^T(\mathrm{x}'_t, a')\boldsymbol{u}^*) \right] - \|\boldsymbol{u} - \boldsymbol{u}^*\|^2,$$

where we used the fact that $F(\boldsymbol{u}^*, \boldsymbol{\lambda}(\boldsymbol{u}^*)) = 0$. Using Jensen's inequality, we get

$$\dot{L}_F(\boldsymbol{u}) \leq \|\boldsymbol{u} - \boldsymbol{u}^*\| \mathbb{E}_\mu \left[ \left\| \gamma \boldsymbol{\phi}(\mathrm{x}_t, \mathrm{a}_t)(\max_{a' \in \mathcal{A}} \boldsymbol{\phi}^T(\mathrm{x}'_t, a')\boldsymbol{u} - \max_{a' \in \mathcal{A}} \boldsymbol{\phi}^T(\mathrm{x}'_t, a')\boldsymbol{u}^*) \right\| \right] - \|\boldsymbol{u} - \boldsymbol{u}^*\|^2$$

$$\leq \gamma \mathbb{E}_\mu \left[ \|\boldsymbol{\phi}(\mathrm{x}_t, \mathrm{a}_t)\| \max_{a' \in \mathcal{A}} \left\| \boldsymbol{\phi}^T(\mathrm{x}'_t, a') \right\| \right] \|\boldsymbol{u} - \boldsymbol{u}^*\|^2 - \|\boldsymbol{u} - \boldsymbol{u}^*\|^2$$

$$\leq \gamma \|\boldsymbol{u} - \boldsymbol{u}^*\|^2 - \|\boldsymbol{u} - \boldsymbol{u}^*\|^2$$

$$\leq (\gamma - 1)\|\boldsymbol{u} - \boldsymbol{u}^*\|^2 \leq 0.$$

The conclusion follows. $\qquad\square$

## C Boundedness of $\boldsymbol{u}_t$ and $\boldsymbol{v}_t$

We conclude by establishing the boundedness of the iterates $\boldsymbol{v}_t$ and $\boldsymbol{u}_t$ under Assumptions (I) through (III). To do that, we use the following result.

**Theorem 6** ([5]). *Given the algorithm*

$$\boldsymbol{w}_{t+1} = \boldsymbol{w}_t + \alpha_t \big( H(\boldsymbol{w}_t) + \mathbf{m}_{t+1} \big),$$

*where*

1. *The function $H : \mathbb{R}^K \to \mathbb{R}^K$ is Lipschitz continuous. Moreover, defining $H_r : \mathbb{R}^K \to \mathbb{R}^K$ as*

$$H_r(\boldsymbol{w}) = \frac{H(r\boldsymbol{w})}{r},$$

*there is a function $H_\infty : \mathbb{R}^K \to \mathbb{R}^K$ such that*

$$\lim_{r \to \infty} H_r(\boldsymbol{w}) = H_\infty(\boldsymbol{w})$$

*for all $\boldsymbol{w} \in \mathbb{R}^K$.*

2. *The origin is a globally asymptotically stable equilibrium of the o.d.e.*

$$\dot{\boldsymbol{w}}_t = H_\infty(\boldsymbol{w}_t).$$

3. *The sequence $\{\mathbf{m}_t, t \in \mathbb{N}\}$ is a martingale difference sequence and verifies, for all $t \geq 0$*

$$\mathbb{E}\left[ \|\mathbf{m}_{t+1}\|^2 \mid \mathcal{F}_t \right] \leq c_0(1 + \|\boldsymbol{w}_t\|^2)$$

*for some $c_0 > 0$.*

4. *The sequence $\{\alpha_t, t \in \mathbb{N}\}$ verifies*

$$\sum_{t=0}^{\infty} \alpha_t = \infty, \qquad\qquad \sum_{t=0}^{\infty} \alpha_t^2 < \infty.$$

*Then, with probability* 1, $\sup_t \|w_t\| < \infty$.

Following [4], we analyze each iterate of the algorithm separately. In particular, we analyze the faster iteration by treating the slower as stationary and analyze the slower iteration by treating the faster as if in equilibrium.

For a fixed $\boldsymbol{u} \in \mathbb{R}^K$, let

$$G_c(\boldsymbol{v}) = \frac{G(\boldsymbol{u}, c\boldsymbol{v})}{c}.$$

We have the following result.

**Lemma 7.** *There is a limiting function* $G_\infty : \mathbb{R}^K \to \mathbb{R}^K$ *such that*

$$\lim_{c \to \infty} G_c(\boldsymbol{v}) = G_\infty(\boldsymbol{v}),$$

*and the origin is an asymptotically stable equilibrium for the o.d.e*

$$\dot{\boldsymbol{v}}_t = G_\infty(\boldsymbol{v}_t).$$

*Proof.* Replacing the definition of $G_c$, we get

$$
\begin{aligned}
\lim_{c \to \infty} G_c(\boldsymbol{v}) &= \lim_{c \to \infty} \frac{G(\boldsymbol{u}, c\boldsymbol{v})}{c} \\
&= \lim_{c \to \infty} \frac{1}{c} \mathbb{E}_\mu \left[ \boldsymbol{\phi}(\mathrm{x}_t, \mathrm{a}_t) \left( \mathrm{r}_t + \gamma \max_{a' \in \mathcal{A}} \boldsymbol{\phi}^T(\mathrm{x}_t', \mathrm{a}_t) \boldsymbol{u} - c \boldsymbol{\phi}^T(\mathrm{x}_t, \mathrm{a}_t) \boldsymbol{v} \right) \right] \\
&= -\boldsymbol{\Sigma}_\mu \boldsymbol{v}.
\end{aligned}
$$

Therefore, letting $G_\infty(\boldsymbol{v}) = -\boldsymbol{\Sigma}_\mu \boldsymbol{v}$, the o.d.e.

$$\dot{\boldsymbol{v}}_t = G_\infty(\boldsymbol{v}_t) = -\boldsymbol{\Sigma}_\mu \boldsymbol{v}.$$

is linear and time-invariant. Since $\boldsymbol{\Sigma}_\mu$ is positive definite, it is immediate that the origin is a globally asymptotically stable equilibrium. □

Noting that all other conditions follow directly from Assumptions (I) through (III) and Propositions 1 through 6, we can now apply Theorem 6 to get the following conclusion.

**Proposition 7.** *Under assumptions (I) through (III),* $\sup_t \|\boldsymbol{v}_t\| < \infty$ *almost surely.*

For the slower iterate, we consider that the faster has converged, and define

$$F_c(\boldsymbol{u}) = \frac{F(c\boldsymbol{u}, \boldsymbol{\lambda}(c\boldsymbol{u}))}{c}.$$

We have the following counterpart to Lemma 7.

**Lemma 8.** *There is a limiting function* $F_\infty : \mathbb{R}^K \to \mathbb{R}^K$ *such that*

$$\lim_{c \to \infty} F_c(\boldsymbol{u}) = F_\infty(\boldsymbol{u}),$$

*and the origin is an asymptotically stable equilibrium for the o.d.e*

$$\dot{\boldsymbol{u}}_t = F_\infty(\boldsymbol{u}_t).$$

*Proof.* Replacing the definition of $F_c$, we get

$$
\begin{aligned}
\lim_{c \to \infty} F_c(\boldsymbol{u}) &= \lim_{c \to \infty} \frac{1}{c} \left( \mathbb{E}_\mu \left[ \boldsymbol{\phi}(\mathrm{x}_t, \mathrm{a}_t) \boldsymbol{\phi}^T(\mathrm{x}_t, \mathrm{a}_t) \right] \boldsymbol{\lambda}(c\boldsymbol{u}) - c\boldsymbol{u} \right) \\
&= \gamma \mathbb{E}_\mu \left[ \boldsymbol{\phi}(\mathrm{x}_t, \mathrm{a}_t) \max_{a' \in \mathcal{A}} \boldsymbol{\phi}^T(\mathrm{x}_t', a') \boldsymbol{u} \right] - \boldsymbol{u}.
\end{aligned}
$$

Let us then define

$$F_\infty(\boldsymbol{u}) = \gamma \mathbb{E}_\mu \left[ \boldsymbol{\phi}(\mathrm{x}_t, \mathrm{a}_t) \max_{a' \in \mathcal{A}} \boldsymbol{\phi}^T(\mathrm{x}_t', a') \boldsymbol{u} \right] - \boldsymbol{u}.$$

Consider the candidate Lyapunov function $L(\boldsymbol{u}) = \frac{1}{2} \|\boldsymbol{u}\|_2^2$. The conclusion of the lemma follows by showing that

1. $L(\boldsymbol{u}) \geq 0$ for all $\boldsymbol{u} \in \mathbb{R}^K$;

2. $L(\boldsymbol{u}) = 0$ if and only if $\boldsymbol{u} = \boldsymbol{0}$;

3. $\dot{L}(\boldsymbol{u}) \leq 0$ for all $\boldsymbol{u} \in \mathbb{R}^K$;

4. $\dot{L}(\boldsymbol{u}) = 0$ if and only if $\boldsymbol{u} = \boldsymbol{0}$.

The first two conditions follow directly from the definition of $L$. As for the last two, we observe that

$$
\begin{aligned}
\dot{L}(\boldsymbol{u}) &= \boldsymbol{u}^T F_\infty(\boldsymbol{u}) \\
&= \boldsymbol{u}^T \left( \gamma \mathbb{E}_\mu \left[ \boldsymbol{\phi}(\mathrm{x}_t, \mathrm{a}_t) \max_{a' \in \mathcal{A}} \boldsymbol{\phi}^T(\mathrm{x}'_t, \mathrm{a}') \boldsymbol{u} \right] - \boldsymbol{u} \right) \\
&= \gamma \boldsymbol{u}^T \mathbb{E}_\mu \left[ \boldsymbol{\phi}(\mathrm{x}_t, \mathrm{a}_t) \max_{a' \in \mathcal{A}} \boldsymbol{\phi}^T(\mathrm{x}'_t, \mathrm{a}') \boldsymbol{u} \right] - \|\boldsymbol{u}\|^2 .
\end{aligned}
$$

Using Jensen's inequality, we get

$$
\begin{aligned}
\dot{L}(\boldsymbol{u}) &\leq \gamma \|\boldsymbol{u}\| \mathbb{E}_\mu \left[ \left\| \boldsymbol{\phi}(\mathrm{x}_t, \mathrm{a}_t) \max_{a' \in \mathcal{A}} \boldsymbol{\phi}^T(\mathrm{x}'_t, \mathrm{a}') \boldsymbol{u} \right\| \right] - \|\boldsymbol{u}\|^2 \\
&\leq \gamma \|\boldsymbol{u}\| \mathbb{E}_\mu \left[ \|\boldsymbol{\phi}(\mathrm{x}_t, \mathrm{a}_t)\| \left| \max_{a' \in \mathcal{A}} \boldsymbol{\phi}^T(\mathrm{x}'_t, \mathrm{a}') \boldsymbol{u} \right| \right] - \|\boldsymbol{u}\|^2 \\
&\leq \gamma \|\boldsymbol{u}\| \mathbb{E}_\mu \left[ \|\boldsymbol{\phi}(\mathrm{x}_t, \mathrm{a}_t)\| \max_{a' \in \mathcal{A}} \left| \boldsymbol{\phi}^T(\mathrm{x}'_t, \mathrm{a}') \boldsymbol{u} \right| \right] - \|\boldsymbol{u}\|^2 \\
&\leq \gamma \|\boldsymbol{u}\| \mathbb{E}_\mu \left[ \|\boldsymbol{\phi}(\mathrm{x}_t, \mathrm{a}_t)\| \max_{a' \in \mathcal{A}} \left\| \boldsymbol{\phi}^T(\mathrm{x}'_t, \mathrm{a}') \right\| \|\boldsymbol{u}\| \right] - \|\boldsymbol{u}\|^2 \\
&\leq (\gamma - 1) \|\boldsymbol{u}\|^2 \leq 0 .
\end{aligned}
$$

The proof is complete. □

Finally, again resorting to Theorem 6, we can state the following proposition.

**Proposition 8.** $\sup_t \|\boldsymbol{u}_t\|_1 < \infty$ *almost surely.*

## Footnotes

[1] Since we are not conditioning on $\mathcal{F}_t$, $\mathbf{v}_t$ is treated as a random variable.

[2] Note that this does not imply, however, that $\sup_t\|\mathbf{v}_t\| < \infty$.