[Reviews · NeurIPS 2020]

Review 1

Summary and Contributions: A new two time-scale approach to Q-learning with function approximation with convergence proof under very general conditions.

Strengths: The authors have a simple algorithm that solves a particular projected Bellman equation. The new problem formulation makes stability analysis as simple as TD learning. This is a remarkable achievement.

Weaknesses: The paper could be in the "award winning category" with a bit more work. They have not thought enough about the flexibility of the formulation. In particular, I believe they can add a matrix in front of u_t in (4a) to ensure that they do not introduce skew in the projection (a concern in section 3.2). The stability analysis is unnecessarily complex at points. Why discuss Lyapunov theory for a linear ODE (for v)? The analysis of the "u ODE" is elegant. It is unfortunate that the results says "we can get a stable algorithm if we are willing to accept bias", and it seems the bias could be huge. Consider the special case of Sigma = sigma I. You can say this is easily achieved by a linear transformation, but then you lose the uniform bound \| \psi (x,a) \| \le 1 . It took me a while to realize how bad this might be.

Correctness: I found a few errors, but nothing fatal: 1. The authors (and many others) claim that (3) is gradient descent on \delta^2. I think this is an old error in the literature. 2. It is unlikely that the stochastic processes on page 5 are martingale difference sequences as claimed. Fortunately, there are now many extensions of the Borkar-Meyn Theorem beyond this special case. Recent papers coauthored by Bhatnagar will do the job. 3. I don't think that the authors have established that their assumptions are weaker than those imposed in [13]. It would be helpful to give an example for which their assumptions hold, and the (very strong) assumptions of [13] fail.

Clarity: Very clear. Some notation is a bit odd. For example, the repeated use of conditional expectation notation (e.g. eqn (5)), when really the authors are simply fixing a parameter. The authors should clarify the proof: make clear that they have a contraction in L_2 and not L_\infty (which is what people expect in Q-learning). I am referring to discussion surrounding eqn (5). Parts of Section 3.2 should be up front -- make clear that the goal is to solve the projected Bellman equation.

Relation to Prior Work: It seems they forgot to reference GQ-learning.

Reproducibility: Yes

Additional Feedback:


Review 2

Summary and Contributions: This paper proposes a variant of DQN with linear function approximation and provides asymptotic convergence analysis using the two time-scale result. The reviewer finds the the strong result is established on a strong assumption.

Strengths: The theory part establishes global convergence result, while the assumptions do not impose any condition on the exploration policy, which is very different from previous work.

Weaknesses: Assumption IV seems to play a critical role in proving the global convergence (Theorem 2), while the explanation of it being weak (line 196-197) is not clear.

Correctness: The proof and the empirical methodology look correct to me, while I did not check all of them.

Clarity: This paper is well written and easy to follow.

Relation to Prior Work: Yes

Reproducibility: Yes

Additional Feedback: For assumption IV, I believe assuming $\Sigma$ is diagonal does not lose generality since we can transfer the feature vector phi using $\Sigma$'s eigenvectors. However, I didn't see why we can assume that the diagonal matrix has equal nonzero elements.


Review 3

Summary and Contributions: This paper presents a variant of Q-learning with convergence guarantees with function approximation. The algorithm, coupled Q-learning, is inspired by DQN and uses two-timescales to assure convergence. Simple experiments support the theoretical guarantees.

Strengths: - The contribution is mainly a novel two-timescale algorithm inspired by DQN with convergence guarantees. The theoretical claims seem to be well-supported and, to my knowledge, this approach has not been taken before. - Empirically, the algorithm also seems to outperform vanilla Q-learning and greedy GQ in certain settings, with greatly improved performance on some simple examples. - I think this contribution would be of interest to the community as another approach to obtain more stable value-learning algorithms.

Weaknesses: - While the theoretical results seem correct, it is not clear to me the advantages of this approach compared to previous work, in particular, gradient Q-learning (GQ). On line 110, it is written that the assumptions are not as stringent but I am not convinced that this is the case. Could the authors clarify this point? - Assumption 1 does not seem very natural to me. If I am interpreting it correctly, it assumes that we have a fixed replay buffer of data on which we are doing updates, as in the offline batch RL setting. It is not specified which policy is used to collect this data and I would expect certain assumptions on this behavior policy. - I do not think assumption 4 would be extremely realistic in practice, but I think this is acceptable in this case since it seems like this assumption is mainly made to provide more insight on the performance of the algorithm (in a special case). As such, I would be more comfortable if it was presented as such, instead of trying to justify it in practice. For example, line 196: "We note that Assumption (IV) does not impose any additional constraint on the features considered, since we can make them orthogonal and scale them to ensure that the latter assumption holds." I do not find this argument convincing since we may be given features that do not satisfy this condition in practice or features may be changing over time if they are being learned. - I would have appreciated some more explanation concerning the intuition of the two-timescale updates. Line 121: "...but instead match the projection of the output along the feature space, much like the “pre-conditioning” step from [1]." seems to hint at the idea but it is not clear to me exactly what this means. - One of the contributions is listed as "A better theoretical understanding for the use of the target network in DQN." (line 112). I think the paper could have included more explanations about this point. Currently, I cannot find much discussion about how CQL informs us about DQN. - For the empirical experiments, it is not clear how the hyperparamters were selected. On line 222, it is simply written what the step sizes are but the algorithms can be sensitive to this choice. It would be much more informative if sweeps over the two step sizes were done and the algorithms' sensitivity to them was assessed. Full parameter sweeps seems feasible in this case due to the simplicity of the environments. - For the mountain car experiments, it is surprising that only 3 training runs were done (line 255) for each algorithm. I would expect many more for such a simple environment, say, at least 10. Currently, the standard errors are very large and it is difficult to make meaningful comparisons between the algorithms.

Correctness: See the section on weaknesses.

Clarity: Yes, the paper was well-organized and easy to follow. There were no obvious typos.

Relation to Prior Work: As mentionned in the "weaknesses" section, from a theoretical standpoint, it is not clear which advantages CQL has over gradient Q-learning (particularly, the assumptions). As another two-timescale algorithm, I think it would be worth discussion more in depth the relation between the two algorithms.

Reproducibility: Yes

Additional Feedback: This is a minor point: One of the motivations of the paper is to try to take inspiration from DQN to produce a convergent Q-learning algorithm. In the end, it seems like the algorithm is quite different from DQN so I am uncertain if CQL gives us much insight into why DQN works. If I am mistaken, then I think it would be better to explain what we learn about DQN through the analysis of CQL. I would be willing to revise my score if the previous points are addressed.


Review 4

Summary and Contributions: The paper introduces a new variant of the classical Q-learning algorithm with linear function approximation with convergence guarantees with less restrictive assumption. The algorithm is designed to include experience replay and a process analogous to the target network to more accurately reflect how reinforcement learning algorithms are often implemented when combined with deep neural networks. Notably, the target network operates at a slower time-scale and its parameters are updated to match the projection of the action-value function onto the feature space. The convergence proof is based on a two time-scale argument based on the ODE approach proposed by Borkar (2008). Finally, the paper presents empirical evaluations in two environments where Q-learning with function approximation has shown to diverge and in the mountain car environment, which is a standard benchmark domain for reinforcement learning algorithms with function approximation. The main contributions of the paper are: 1. Algorithmic: The paper proposes a new algorithm, Coupled Q-Learning or CQL, based on the Q-learning algorithm. The design of this algorithm accounts for architectural additions often used in deep reinforcement learning, i.e., the replay buffer and the target networks. 2. Theoretical: The paper presents convergence results for the proposed algorithm. Specifically, the paper proves that the processes generating the parameters of each of the two sets of weights reach an equilibrium point and provides a bound between the approximate and optimal action-value functions. 3. Empirical: The paper presents empirical evaluations on two environments where Q-learning is known to diverge when combined with linear function approximation and on mountain car, a standard benchmark for action-value methods with function approximation. References: V. Borkar. Stochastic Approximation: A Dynamical Systems Viewpoint. Cambridge University Press, 2008.

Strengths: The ideas presented in the paper excel in their novelty, theoretical soundness, and significance. The novelty of the algorithm stems from the inclusion of two common techniques often used in deep reinforcement learning — experience replay and target networks — and providing theoretical guarantees that take those techniques into account. The theory and the proofs are carefully done and intuitively explained in the main text and the appendix. Finally, the results in the paper are significant because they provide a justification for the use of such techniques and demonstrate that these techniques are useful for overcoming the deadly triad problem encountered when combining bootstrapping, function approximation, and off-policy learning in reinforcement learning. I would expect these results to be very significant for the whole community studying reinforcement learning.

Weaknesses: The only weakness of the paper lies in its empirical evaluations. The first two empirical evaluations — the theta to two theta environment and the star counterexample — are impressive, but they could further be improved by providing some explanation about the choice of hyperparameters. It is clear that CQL converges in both environments; however, it would be useful to get a sense of how sensitive CQL is to its two learning rates and also to know whether GGQ was given a fair shot. For the third experiment in the mountain car environment, the motivation is not as strong as for the first two experiments. The only motivation for this experiment is to test CQL in a more complex environment. However, I would argue that the results of this experiment could be presented in a way that compliment the results of the first two experiments, which would make the motivation stronger. In the first two experiments, the motivation is to test the convergence properties of CQL in environments where Q-learning is known to diverge. The hypothesis of the experiment is that, unlike Q-learning, the approximate action-value function computed by CQL will converge to zero. The motivation for the third experiment would then be to test the performance of CQL in an environment where Q-learning doesn’t face any convergence issues. The hypothesis would then be that CQL will have similar performance to Q-learning. The reason why I think this would make for a stronger argument in the paper is that the first two experiments would demonstrate that CQL has stronger convergence properties than Q-learning, whereas the third experiment would demonstrate these improved convergence properties don’t result in a great loss of performance. Together this would demonstrate that there are significant gains in using the CQL algorithm. Nevertheless, these changes would require the redesign of the third experiment. I would suggest to provide a more detailed view of the performance of each algorithm by presenting learning curves that show their behaviour during learning. As for how to select hyperparameters, there are two options. One option would be to do an extensive hyperparameter sweep and then present the results corresponding to the best parameter combination. This is standard, but not necessarily the best. A stronger argument could be made by using a more systematic evaluation such as the one presented in Jordan et. al. (2020) paper on performance evaluation in reinforcement learning. This second alternative would provide a more complete comparison between the performance of Q-learning and CQL. Finally, is there a reason why the third experiment uses radial basis functions? I’m mostly concerned that the results showed that most algorithms failed to learn most of the time which seems indicative of a poor feature construction function. I would suggest using Tile Coding (Sutton and Barto, 2018, also see http://incompleteideas.net/tiles/tiles3.html for an implementation) since it often results in good performance in the mountain car environment (Sutton, 1996). References: Jordan, S., Chandak, Y., Cohen, D., Zhang, M., & Thomas, P. (2020). Evaluating the performance of reinforcement learning algorithms, In Proceedings of machine learning and systems 2020. Sutton, R. S., & Barto, A. G. (2018). Reinforcement learning: An introduction. Cambridge,MA, USA, A Bradford Book. Sutton, R. S. (1996). Generalization in reinforcement learning: Successful examples using sparse coarse coding. In Advances in Neural Information Processing Systems 8 (NIPS 1995), 512 pp. 1038–1044. MIT Press, Cambridge, MA.

Correctness: Couldn’t find any mistakes in the proofs.

Clarity: Overall, the paper is well written. I found two typos: - Line 38: the in-text citation of Mnih et. al.’s paper is missing the year in parenthesis (2015). - Line 118: there’s an equal sign before the << in the expression describing the relationship between the two learning rates. I don’t think it should be there. For clarity, it would be useful to mention that the norm considered throughout the main text of the paper is the L2 norm. Also, Line 159 should mention in plain text that \mathcal{ F }_t is a sigma-algebra. In Line 34, write Asynchronous Dynamic Programming instead of ADP. The term is not introduced before that point in the paper, so it is not immediately clear what ADP is.

Relation to Prior Work: The literature review is thorough and the connection to previous work and how the current work differs from it is clear.

Reproducibility: Yes

Additional Feedback: How sensitive is CQL to its two learning rates? I would consider increasing my score if the authors addressed my suggestions about the mountain car experiment. === Final Comments === After the authors rebuttal and discussion period, I consider the paper to be marginally above the acceptance threshold. I think it would make a stronger submission if the paper included a more in-depth discussion about the bias of CQL in a more general case.

[Author Response · NeurIPS 2020]

**Reviewer 1:** We thank the reviewer for the thoughtful questions and comments, which we address below.

○ Adding a matrix to $u_t$ can, in fact, help with the skewness in the projection. The obvious choice is $\phi(x_t, a_t)\phi^T(x_t, a_t)$, but the resulting algorithm does not retain the convergence guarantees. In that case, the convergence analysis presents the same difficulties of standard $Q$-learning. We also explored adding the matrix $\text{diag}(\phi(x_t, a_t))$, resulting in an algorithm that has tabular $Q$-learning as a particular case, concerning the provided limit solution. Again, convergence does not hold so generally.

○ We appreciate the reviewer's suggestion that the stability analysis of the fast o.d.e. (for $v$) can be simplified and will do so in the final version of the document. We also agree that the original $Q$-learning with function approximation (3) is not true gradient descent on the Bellman error and will correct the statement in the final version of the document.

○ Regarding the martingale difference sequences, we believe that our proof (Appendix B.3) establishes this assertion from Assumption (I). Nevertheless, we thank the suggestion of using Bhatnagar's recent works, which could provide an elegant alternative proof.

○ We break down the comparison between our assumptions and the ones from [13] in three, namely (i) technical assumptions; (ii) assumptions on the data; (iii) assumptions on the features. (i) Both works use standard conditions to facilitate the convergence analysis; (ii) [13] requires uniform ergodicity of the sampling policy, which is stronger than our assumption; (iii) while [13] requires features such that, in terms of $\Sigma_\mu = E_\mu[\phi\phi^T]$, the sampling policy is very close to the optimal policy, our assumption is met through normalization alone. For example, the assumptions of [13] do not hold on the $\theta \to 2\theta$ example in Section 4, contrarily to ours.

**Reviewer 2:** We thank the reviewer for the thoughtful questions and comments, which we address below.

We start by clarifying that Theorem 2 is not a convergence result, but rather an error bound. We will make this explicit in the final version of the document. Concerning the generality of Assumption (IV), particularly why we can assume that $\Sigma_\mu = E_\mu[\phi\phi^T]$ has equal non-zero elements in the diagonal, we note that Assumption (II) requires $\Sigma_\mu$ to be non-singular, which is equivalent to linear independence of the features. As such, we can assume that the features are orthogonal without loss of generality.[1] Each element in the diagonal of $\Sigma_\mu$ is thus the norm of the corresponding feature, in the norm induced by the inner product $\langle f, g \rangle = E_\mu[f^T g]$. We can thus scale all features appropriately to ensure (IV).

**Reviewer 3:** We thank the reviewer for the many thoughtful questions and comments, which we address below.

○ Regarding the advantages of our approach in comparison with Gradient $Q$-learning (greedy-GQ, [12]), our approach converges to a unique, well-defined solution that is independent of the initialization, whereas GQ may converge to any local optimum (see discussion in Section 1) and our assumptions are similar. If the reviewer refered instead to Gradient $Q(\sigma, \lambda)$ (GQ$(\sigma, \lambda)$: A Unified Algorithm with Function Approximation for Reinforcement Learning), we notice, for instance, that their convergence result is established assuming the iterates remain bounded (Assumption 2(3) of their work), which we prove to be true in our case (Appendix C).

○ Although technically Assumptions (I), (II) and (III) are sufficient for convergence, the reviewer is right in that the policy used to collect the data is fundamental, since the limit solution directly depends on it through the distribution $\mu$.

○ We acknowledge and will incorporate the clearer introduction to Assumption (IV) suggested by the reviewer.

○ The use of a two time-scale algorithm replicates the update structure of DQN: in DQN, a target network is mostly fixed and updated only infrequently; in CQL, the target network is updated on every time step, but very slowly (Section 2.1). The two time-scale formulation thus "mimics" the dynamics of the target and main networks in DQN, while being amenable to analysis using results from the stochastic approximation literature.

○ CQL builds directly on two key elements of DQN: experience replay (Assumption (I)) and a target network ($u$). Although the actual architecture of the two approaches is distinct (ours is linear, while DQN is non-linear; the target network is also not exactly the same), we still believe that our analysis of CQL provides theoretical insight regarding the two aforementioned elements of DQN.

○ Finally, regarding the experiments, we did perform an empirical sensitivity analysis to the learning rates, which showed CQL is robust to such variations. We can include those results in the supplementary material. Also, in the mountain car experiment we actually performed 10 runs, not 3. We thank the reviewer for bringing to our attention that this is not clearly phrased in the paper.

**Reviewer 4:** We gratefully acknowledge the reviewer's comments and suggestions, particularly with respect to the experimental section. We will incorporate these into the manuscript, since we agree that these help to make a clearer experimental section (namely with respect to the mountain car experiment).

○ Regarding the choice of parameters, we refer to our rebuttal to Reviewer 3. Adding to that, the choice of learning rates will be formalized through the complete data collection method, from the work the reviewer fittingly suggested (Jordan et al. (2020)).

○ We agree that the motivation for mountain car is inevitably weaker, given the focus of our work on convergence issues. We acknowledge the suggestion of presenting more detailed performance metrics, such as learning curves. We can add these to the supplementary material and, to the best of our ability and within space limitations, to the main paper.

○ Regarding the use of radial features, we use them out of their simplicity and natural compliance to Assumption (IV). On one hand, if their radius is small and the data is sampled uniform over states and actions, Assumption (IV) immediately follows. On the other hand, as the radius increases, the "violation" of Assumption (IV) increases accordingly, which facilitates assessing the impact of Assumption (IV) on the performance of CQL.

## Footnotes

[1] If not, we can use Gram-Schmidt's method to obtain equivalent orthogonal features.


[Meta-Review · NeurIPS 2020]

This paper presents a new objective and an algorithm, which is similar to DQN, that optimises for that objective. Similar to prior work (GTD, TDC, GQ), the algorithm is shown to be convergent under linear function approximation. Because the objective is different, the paper could have better illustrated what this means in terms of the quality of the fixed point the new algorithm converges to - this is only discussed in detail in a special case of a diagonal feature covariance matrix. The author response did not lift this concern, and it remains unclear whether the new algorithm has major benefits over existing related work. The experiments were deemed somewhat insufficient to fully convince the reviewers of this. Overall the reviewers found the paper to be interesting, and therefore recommend (though non-unanimously) to accept the paper. The authors are encouraged to provide more understanding and/or validation in the final version of the paper.